palaeontology/ecology

transoceanic rafting, Holzmaden lagerstätte, spatial analysis, diffusion analyses, Crinoidea

**Author for correspondence:**
Aaron W. Hunter
e-mail: awh31@cam.ac.uk

# Reconstructing the ecology of a Jurassic pseudoplanktonic raft colony

Aaron W. Hunter[1,2], David Casenove[3], Celia Mayers[4] and Emily G. Mitchell[5]

[1]Department of Earth Sciences, University of Cambridge, Downing Street, Cambridge CB2 3EQ, UK
[2]School of Earth Sciences, University of Western Australia, 35 Stirling Highway, Crawley, WA 6009, Australia
[3]Graduate School of Nursing, Chiba University, 1-8-1, Inohana, Chuo-ku, Chiba-shi, Chiba 260-8672, Japan
[4]Department of Applied Geology, Curtin University, Perth, Western Australia, Australia
[5]Department of Zoology, University of Cambridge, Downing Street, Cambridge CB2 3EJ, UK

AWH, 0000-0002-3527-5835; DC, 0000-0003-1377-7030;
EGM, 0000-0001-6517-2231

Pseudoplanktonic crinoid raft colonies are an enigma of the Jurassic. These raft colonies are thought to have developed as floating filter-feeding communities due to an exceptionally rich oceanic niche, high in the water column enabling them to reach large densities on these log rafts. However, this pseudoplanktonic hypothesis has not been quantitatively tested, and there remains some doubt that this mode of life was possible. The ecological structure of the crinoid colony is resolved using spatial point process analyses and the duration estimates of the floating system until sinking using moisture diffusion models. Using spatial analysis, we found that the crinoids would have trailed preferentially positioned at the back of the floating log in the regions of least resistance, consistent with a floating, not benthic ecology. Additionally, we found using a series of moisture diffusion models at different log densities and sizes that ecosystem collapse did not take place solely due to colonies becoming overladen as previously assumed. Our analyses have found that these crinoid colonies studied could have existed for more than 10 years, even up to 20 years, exceeding the life expectancy of modern documented raft systems with possible implications for the role of modern raft communities in the biotic colonization of oceanic islands and intercontinental dispersal of marine and terrestrial species.

## 1. Introduction

Transoceanic rafting is a fundamental feature of marine evolutionary biogeography and ecology, which is used to explain the origins of

**Figure 1.** Crinoid fossil raft, the 'Hauff specimen' from Holzmaden (G1). (*a*) Log with spatial analysis data points; key: blue = crinoid crowns, black = attachment discs. (*b*) Spatial analysis plot. (*c*) Close up view of the crinoid crown and stem sections. (*d*) PCF distance plot.

modern global patterns of species distributions [1–4]. Extant communities have been recorded lasting up to 6 years [5]. However, the deep time ecology of these communities has never been investigated using the latest quantitative methods to test these different hypotheses [6] empirically. In recent communities, such rafts have included highly adapted bivalves, barnacles, limpets, bryozoans, sea anemones, amphipods and isopods [5]. In the Jurassic, these communities also consisted of specially adapted crinoids, whose apparent maturity suggests that these pseudoplanktonic communities had to have lasted longer than modern examples (more than 6 years) [6]. The structure and duration and these colonies have remained unquantified, with most studies focusing on crinoids' pseudoplanktonic adaptions and their lifestyle [7,8] rather than the viability of the system, the ecological structure and longevity of the habitat. This study uses the latest quantitative ecological techniques used in palaeobiology to reconstruct the ecology and duration of these rafts.

Crinoids or sea lilies were a major part of the Jurassic shallow sea ecosystem, with crinoids found in a diverse suite of shallow marine environments [9]. The monospecific crinoid colonies preserved on wood rafts are one of the most enigmatic and iconic of these communities [10]. They existed during the Toarcian Oceanic Anoxic Event, a brief period of geographically widespread dysoxic conditions [11]. Found globally, they are also the only fossil example of transoceanic rafting with up to 100 individuals covering bivalve-encrusted logs up to 14 m long [12]. It has been debated whether these crinoids could have colonized and persisted on these floating-log habitats, or they were instead part of benthic islands systems typical of the Mesozoic [13]. Recently, the pseudoplanktonic hypothesis has become widely accepted mainly because (i) these crinoids are always preserved associated with large driftwood, (ii) they are found in anoxic sediments, and (iii) they possess morphological features critical for a pseudoplanktonic lifestyle (long stem enabling efficient tow-net filtration). Although previous studies have qualitatively examined the distribution of crinoids on the logs, their distributions were not quantitatively described [14,15]. In this study, we use spatial point process analyses (SPPA) and diffusion modelling in a novel combination to test whether this pseudoplanktonic mode of life existed. SPPA was used to test whether the logs were colonized in open water or on the substrate, and diffusion models are then used to quantify floating-log mechanics to test how long a floating system could have existed.

The spatial positions of *Seirocrinus* on one of the largest and best-preserved Early Jurassic floating wood examples known, the large 'Hauff Specimen' from Holzmaden, Germany [15] were mapped following Mitchell *et al.* [16] (figure 1; electronic supplementary material, figures S1–S3). The spatial

patterns of benthic organisms depend on the dispersal of larvae [17,18], the environmental conditions in which they settled [16] and whether the conditions were favourable for them to grow to adulthood [19,20]. Therefore, the spatial patterns of the crinoids can be used to try to 'reverse engineer' the conditions in which they settled in order to deduce the environmental conditions (what part of the water column the log was in) when colonized (cf. [21,22]).

This analysis was complemented by analyses using density/diffusion models to constrain the length of time logs of differing sizes and densities, with a full colony, could stay afloat as water infiltrates the wood over time [23,24]. This diffusion analysis used three sets of size-defined colonies (electronic supplementary material, figures S4–S9), including small colony specimens (S1 and S2) (electronic supplementary material, figures S4 and S5), medium colonies (M1 and M2) (electronic supplementary material, figures S6 and S7) and large colonies from Holzmaden (G1 'the Hauff specimen' and G2) (electronic supplementary material, figures S8 and S9).

# 2. Material and methods

## 2.1. Data collection

Photographs (Nikon D5500 SLR Camera) were taken from six specimens, and composite images produced using Adobe Illustrator. These include small, medium and, finally, large colonies from the following German museums and collections: Geoscience Centre of the University of Göttingen; Werkforum Museum, Dotternhausen; Naturmuseum Senckenberg in Frankfurt am Main; Geological Institute, University of Tübingen; Staatliches Museum für Naturkunde, Stuttgart; and Urweltmuseum Hauff, Holzmaden.

## 2.2. Spatial analysis

Randomly distributed points (figure 1a) can be modelled using homogeneous Poisson processes and are found, for example, when larvae attach to a substrate not impacted by strong currents nor patchy environmental variables [25]. Spatial processes that depend on position on the substrate can be modelled using heterogeneous Poisson processes, with underlying density changing according to environmental conditions such as patchiness of a substrate [26], or other physical conditions represented whereby the density is modelled by a formula. Disc density was modelled as a heterogeneous Poisson process dependent on the $x$-coordinate and then the $y$-coordinate to assess how disc density changed along the log. Model fit was assessed using the model residuals by plotting Q–Q and smoothed residual plots. If the observed line in the Q–Q plot fell outside two standard deviations of the model, the model was rejected [21,27]. Akaike information criterion (AIC) values [28] were used to compare the relative quality of the statistical models that fitted the data.

## 2.3. Sinking log models

As it is not possible to know the actual properties of the fossil wood (due to the lack of fossil examples available for study) or the actual growth rate of the byssate inoceramids on such substrate (as *Pseudomytiloides dubius* is now extinct), two types of models were developed. The first focuses on the soaking process, which estimates the potential duration for a wooden log to stay afloat as a function of the size and porosity of the wood material. This model takes into account the microstructure and density of the wood. For example, dense woods will take longer to absorb water than less dense ones, and some wood types have transverse struts in the vessels called tyloses, which slow waterlogging. The model is not able to test the osmotic pressure from seawater that aids buoyancy or a process called bacterial sealing, whereby bacterial slime clogs vessels within the wood [19]. These processes will increase the viability of the system and the length of time the log colony would have lasted. The second model examines the impact of the growth of a community of *P. dubius* (Sowerby) on the buoyancy of wooden logs using fossil data about abundances, size distributions and individual growth patterns in the literature. In this second model, logs are considered sealed, which means that the weight increase of a log is solely due to the growth of the bivalve community that has settled on it. Due to the lack of extant representatives of *P. dubius*, uncertainties remain concerning the seasonal variations of its growth or the biological parameters related to its population dynamics.

### 2.3.1. Model 1: diffusion analyses

A Fickian diffusion model was used to evaluate the soaking time of wood logs [23,24]. The debate is still open concerning the Fickian nature of the moisture transport as water appears in three distinct phases in wood: free, bound and gaseous [29]. However, as there is currently no model suited to the non-Fickian behaviour in wood [30], we decided to use a Fickian model to provide an order of magnitude for the soaking durations until sinking. Despite being extremely simplified, it has been shown to be equivalent to other models, at least in isothermal conditions [29]. Previous research has shown that the Fickian model systematically underestimates the value of diffusion coefficients mostly during the initial phase of absorption [31] which means that it predicts soaking speeds that are slower than the experimental measurements. Therefore, the estimated times to sinking considered in this study should be considered as maximum values.

Wood is an anisotropic material composed of dead and living cells through which moisture is transported (see electronic supplementary material, S2.1 Physical properties of wood). Due to the hygroscopic nature of this material, the physical properties of wood, including its moisture transport properties, are strongly influenced by both the surrounding moisture as well as the ambient temperature [32]. In addition, the structure of the wood itself and the agents of decay in the environment the wood is found are critical for the longevity of the system and how the log structure will break up (see electronic supplementary material, S2.1.4 Decay).

Moisture content in a wood is the percentage of the weight of the water as compared to its oven-dry weight (values beyond 100% are common). This model assumed that the wood was green with a moisture content higher than the fibre saturation point (greater than 35% moisture content) to avoid any impact of volume change for the vessels. We also assumed that the wood is constantly at equilibrium with the atmosphere. As the free water is already in the vessels, most of the soaking is the result of the diffusion of either water bound to the cell walls (bound water) or vapour. The present two-phase model examined the diffusion of moisture under isothermal conditions using a finite difference technique to solve the differential equations that involve diffusion coefficients that were varying according to the moisture content. The model considered an isothermal diffusion because heat diffusivity is orders of magnitude higher than moisture diffusivity in wood. Three directions were considered when examining tree wood: the longitudinal, radial and tangential directions. The longitudinal direction is the direction of the wood fibres. The radial direction runs from the centre of the tree to the outside. Finally, the tangential direction follows the curvature of the tree [23]. For simulations, we considered that radial water transfer encompasses both radial and tangential diffusion processes. Longitudinal water transfer can be expressed as a function of the density of the material as well as the temperature and moisture content of the material (see electronic supplementary material, S2.2 Moisture sorption model, equation 12). Radial water transfer is more difficult to evaluate because of structural variation in the wood fibres/vessels, but experimental data suggest that the radial diffusion coefficient of moisture ($Dr$) can range between 10 and 1000 times less than the value of the longitudinal diffusion coefficient ($Dl$) for moisture contents beyond the saturation point (see electronic supplementary material, S2.2 Moisture sorption model, equation 22). Details for the mathematical model of diffusion (calculation of the diffusion coefficients for all water phases and geometry of the diffusion) are available in the electronic supplementary material (see electronic supplementary material, S2.2.1–2.2.4).

Finally, we considered the impact of a crinoid community that grew on the surface of our modelled logs by assuming all the crinoids found at the surface of the log settled at the start of the simulation and grew regularly during the soaking phase (see electronic supplementary material, S2.2.5). The log was loaded with a community of crinoids with an arbitrary individual growth rate of $8\,\mathrm{cm\,yr^{-1}}$ extrapolated from extant species of *Neocrinus decorus*, *Endoxocrinus parae* and *Cenocrinus asterius* within $(0.6–17.0\,\mathrm{cm\,yr^{-1}})$ [24].

### 2.3.2. Model 2: *Pseudomytiloides dubius* community growth model

The second model considers the growth of the inoceramid bivalve community on the surface of the same logs and estimates how long the log can remain afloat while bearing the weight of the colony. It relies on data from the literature about *P. dubius* [33]. The model of population growth assumes the settlement on one single individual when the log contacts seawater and estimates the added mass of the community to the wholly sealed log every 1-year increment until the whole log and colony sinks due to its density (see electronic supplementary material, tables S2 and S3).

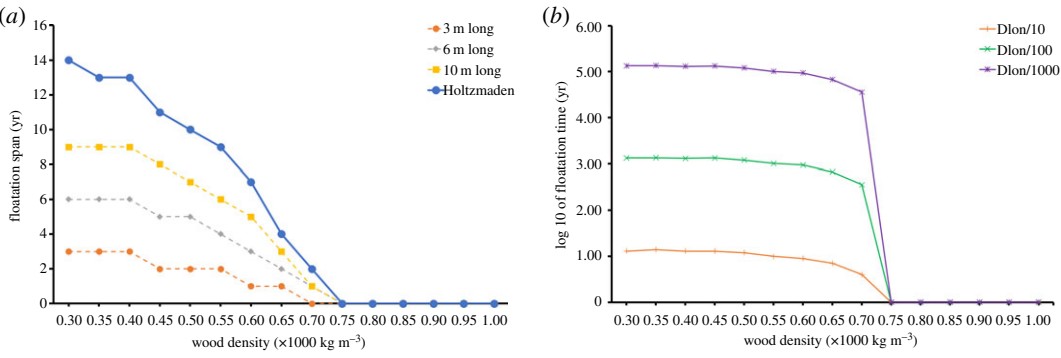

**Figure 2.** Estimates for the soaking duration (in years) of cylindrical log due to (*a*) longitudinal and (*b*) radial diffusion of water. Wood logs can stay afloat several years before absorbing enough water to reach a critical density allowing them to sink. Modelled values on the vertical axis indicate the number of years required for three cylindrical logs to sink and are presented in relation to the length and density of the material. For each geometry, durations were calculated for three distinct values of radial diffusion coefficient (Dlong/Drad = 10; Dlong/Drad = 100 and Dlong/Drad = 1000). In all cases, the wood log was assumed to have reached an equilibrium moisture content (EMC) with an atmosphere at 20°C and 100% humidity. In such conditions, wood densities above $0.75 \times 10^3$ kg m$^{-3}$ will sink instantly.

## 3. Results

The spatial analyses show directionality significant for the positions of crinoid attachments along the log with the highest density on the left-hand side of the figure, with decreasing density linearly along the log (figure 1*b*; electronic supplementary material, figures S2 and S3). This directionality can be modelled by a heterogeneous Poisson model of a linear model using depending on the *x*-coordinate with AIC model comparisons resolving this model over other, more complex ones (electronic supplementary material, table S1). The diffusion analysis is represented by two models, which show that the largest of the log systems (Holzmaden) could have survived for a minimum of 2 years and a maximum of 14 years (20 years if we account for the maximum values of the longitudinal diffusion coefficient) with an atmospheric relative humidity of 100% (figure 2; electronic supplementary material, figures S4–S9) thus allowing the crinoid colony to grow to maturity and confirming the viability of the pseudoplanktonic hypothesis. Longer soaking durations of up to 60 years have been calculated for cases where the wood dried prior to enter in contact with water (figure 3, electronic supplementary material, figures S4–S9). When the diffusion model was applied to the small logs (S1 and S2) (figure 3*a* and *b*, electronic supplementary material, figures S4 and S5) without the crinoid population, it showed that the log could have stayed afloat for 800 months; however, when the colony was added it sank after one time increment (one month) (figure 3*b*, electronic supplementary material, figure S5). The apparent longevity of the logs is increased by their size, with the medium-sized logs (M1 and M2) (electronic supplementary material, figures S6 and S7) surviving 100 months (800 without the colony) and the large colonies (G1 and G2) (electronic supplementary material, figures S8 and S9) surviving up to 400 months (800 without the colony) (electronic supplementary material, figure S9). Our results suggest that without any natural sealing, the largest rafts sizing up to 12 m could have stayed afloat for a maximum theoretical duration of 2–20 years with a surrounding humidity of 100%, with the smaller systems unviable despite being relatively common. Due to the design of the model, it is likely that these values overestimate the actual duration of the diffusion process.

The bivalve population model incorporated both the spatial distribution along the log as well as life-history estimates (fecundity, mortality, settling rates and maturation time) for a complete life cycle using extant bivalve representatives in the Mytiloida (*Mytilus edulis*) and the Ostreoida (*Ostrea chilensis*), as modern analogues to the byssate inoceramids (Pteriomorphia, Pteriida, Ambonychioidea, Inoceramidae) found at Holzmaden. The model allowed the reconstruction of the number of years required to reach sinking density for the whole log for various values of carrying capacity K (electronic supplementary material, tables S2 and S3). If we consider the geometry of the Holzmaden log, the population model suggests that, without any simultaneous absorption of water, the bivalve community could grow for more than 10 years before reaching a critical mass causing the log to sink. In some cases, the dimensions of the log combined with the density of the wood material would allow the bivalve-loaded log never to reach the sinking mass threshold within a 20-year window (electronic supplementary material, table S3).

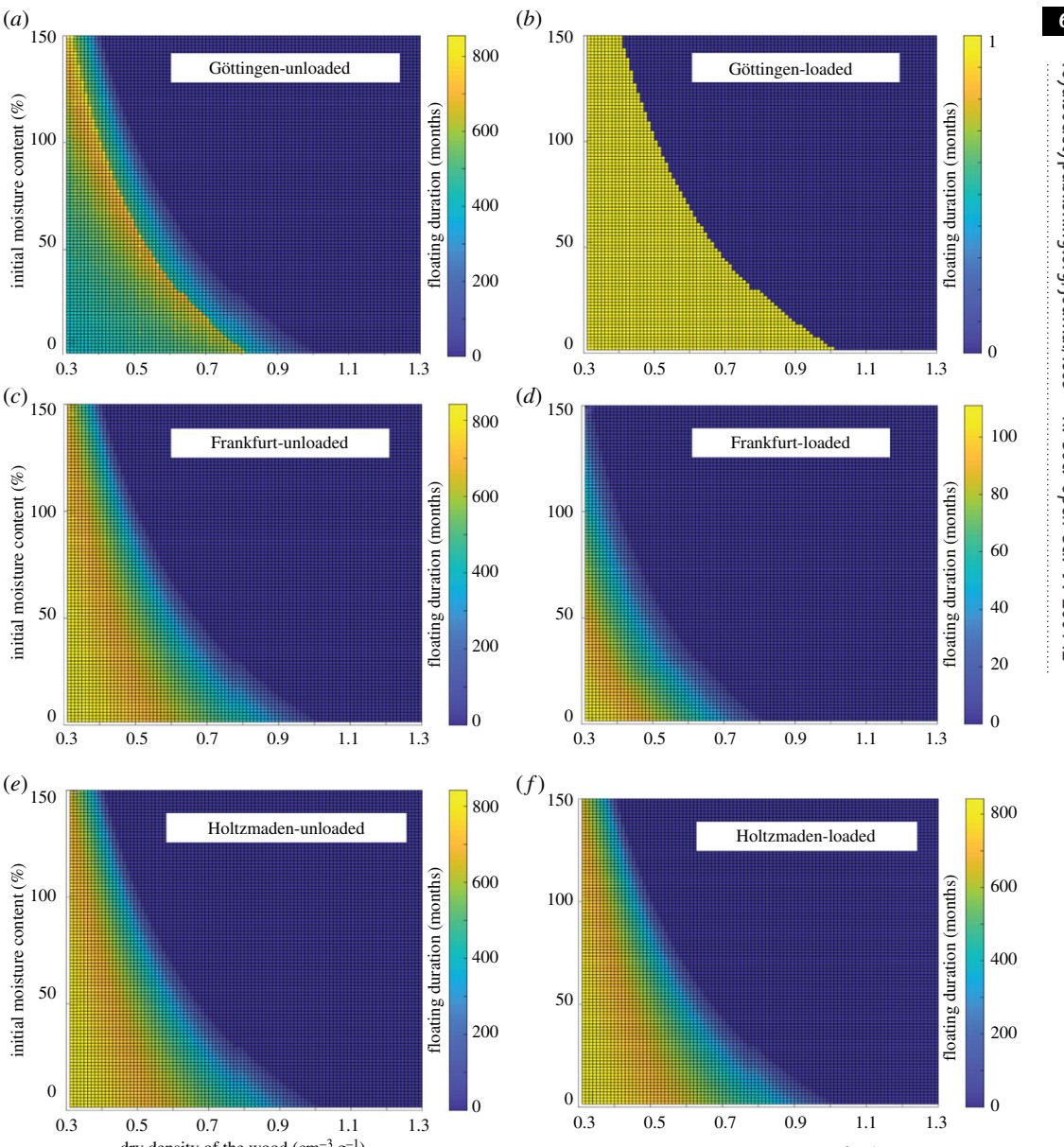

**Figure 3.** Duration of the floating phase (in months) for varying density (horizontal axis) and varying equilibrium moisture content (EMC). Calculations are run for three specimens: Gottingen (S1), Frankfurt (M1) and Holtzmaden (G1). (*a*) S1 log sample with no crinoid population (unloaded). (*b*) S1 log sample with crinoid community (loaded). (*c*) M1 unloaded. (*d*) M1 log sample loaded. (*e*) G1 log sample unloaded. (*f*) G1 log sample loaded.

## 4. Discussion

Our findings suggest that a pseudoplanktonic lifestyle was possible for larger logs but not for smaller specimens. The crinoids attached preferentially to one end of the log structure (figure 1*b*; electronic supplementary material, figures S2 and S3), with a linear decrease of density down the log, and attachment discs barely developed on the upper surface (figure 3). This result confirming existing observations was also made in an exceptionally preserved example from Holzmaden [8] and supports the hypothesis that our largest colony was structured as a floating body, not a benthic structure. Spatial distribution is notable in that it differs from all other previous palaeocommunities SPPA analyses of Ediacaran benthic communities across a range of different environmental settings [16,18,20,22,26,28]. There are no benthic palaeocommunities which show this linear structure. This feature is demonstrated by modern shipworms, for example, which preferentially accumulate at one end of the structure [34].

In the case of a ship, this would be the back or stern. If the log was passively moving through the water and had a preferential orientation, the bow of the structure would still be subject to the most current pressure, so the most hospitable part of the structure would be the area of least resistance, the stern. However, this asymmetric distribution, combined with the observation that the vast majority of crinoids are attached to the underside of the log, the top side is exposed with either smaller individuals or undeveloped attachment discs (figures 1*b* and 3; electronic supplementary material, figures S2 and S3), would make the log unbalanced towards one end, bringing the colony's stability into question. After the settlement of the attachment discs, these fixed crinoids would then be subject to a 'struggle for survival' that would influence their distribution and favour those individuals in the area of least resistance, the stern. The conventional view is that the community would have rapidly sunk and been preserved at anoxic depths due to becoming unstable or overladen [15].

However, our diffusion model does not support rapid sinking for two key reasons. First, the growing community of bivalves and crinoids, even at the climax state, would still be at most 50% of the total weight of the system. This total maximum weight (in exceptional cases) would have varied from 630 to 880 kg depending on carrying capacity compared to total log mass estimates of between 1100 and 1500 kg depending on the wood type (electronic supplementary material, S2.2.5 Weight of the crinoid colony). Our models suggest the Holzmaden log could have supported the population of crinoids for a substantial amount of time. Even with a system designed to be a catastrophic scenario taking on water at a constant rate until the log raft system failed, the largest examples (G1 and G2) (figure 3*e* and *f*, electronic supplementary material, figures S8 and S9) could have survived for up to 2–20 years. Preserved fossil examples in this study were close to their maximum point viability as a mobile substrate. This point is evident from smaller logs (S1 and S2) (figure 3, electronic supplementary material, figures S4 and S5) whose population was simply too big to be supported by the smaller log structure. However, the model looks at the viability of the climax community that would take time to develop, and would expand the life of the colony within the parameters of the model more than 2 years. Modern observed growth rates for recent isocrinid crinoids 30–40 cm [35] in length suggests it would have taken at least 10 years for *Seirocrinus* to reach maturity. Although the growth rates of this crinoid may have been faster due to the unusual structure of the stem [36]. For these communities to exist to maturity the log needs to float for much longer than 2 years. Our bivalve community growth model, in contrast, predicts that the raft was viable for several years and would never reach a critical mass within 20 years for some values of wood density and carrying capacity (electronic supplementary material, table S3).

The key to explaining the prospects of the raft lifespan is firstly by identifying areas of weaknesses where the structure is most likely to fail (such as breaks in the bark), and secondly the ends of the log system that expose the wood xylem [37]. However, in all the examples examined, most of the surface is covered in bivalves (figure 1), and our results suggest that the crinoids themselves are preferentially distributed at one terminal end of the system (figures 1*b* and 4; electronic supplementary material, figures S2 and S3), contributing to maintaining the integrity of the log system. Not all areas of the log were covered in crinoids or bivalves, and this would have shortened the lifespan of the raft system.

A second key that would influence the viably of the community would be the natural properties of the wood itself. Although it is hard to model the wood structure with no fossil remains available in the Posidonia Shale [38], a gymnosperm wood structure with narrow tracheids could have significantly slower moisture diffusion rates than an angiosperm with wider vessels [39]. Estimates vary for how long modern gymnosperm pine driftwood can survive (see electronic supplementary material, S2.1.4 Wood decay). Some estimates are as little as 17 months [40], while others suggest that the wood could have remained buoyant up to 5 years after entering the marine environment [5]. Considering the exceptional properties of the Jurassic gymnosperm wood (with much smaller pore spaces), combined with a possible natural coating that maintains the mechanical integrity of the log, and in the absence of modern aggressive wood-boring predators, which evolved later in the Mesozoic [41], the wood would have probably floated for longer than the 5 years observed in today's driftwood. The colony could, therefore, have survived for more than 10 years needed for a community to reach maturity until the decay and water infiltration of the wood reached a limit where the log would sink. At which point, the log could no longer support any further community growth, and the crinoids died, being perfectly preserved by the anoxic conditions on the seabed [12] (figure 1*a,c*). Evidence suggests that fragments might have broken off while the system was in a state of decay, which might explain the occurrence of single mature individuals attached to relatively small fragments of wood observed across the museum collections examined. This hypothesis could support our findings that smaller logs could not have been buoyant enough to carry the crinoid specimens.

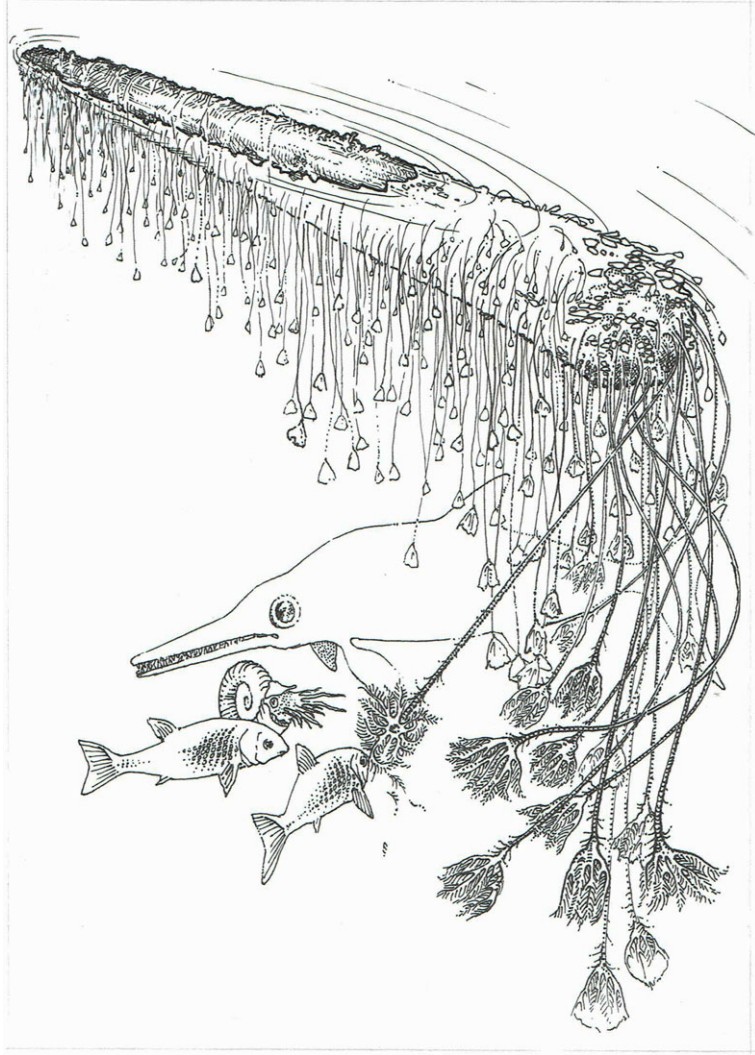

**Figure 4.** Reconstruction of the Holzmaden 'Hauff specimen' crinoid raft colony (G1).

Pseudoplanktonic rafts are highly significant for global ecology and biodiversity, as a mechanism for global colonization [42], typically attaching themselves to nektonic and planktonic organisms or other floating objects such as driftwood or flotsam. These rafts are significant in today's ecosystems and were responsible for the colonization of oceanic islands such as Hawaii [43]. These temporary rafts can become a permanent home to self-contained long-term communities such as, in modern ecosystems, external goose barnacles, tunicates and bryozoans [44], with wood-boring bivalves such as shipworms inhabiting the internal structures [45].

Our analyses suggest that the exceptionally preserved Jurassic crinoid rafts are the longest surviving temporal pelagic rafting communities to exist in the fossil record. Our results demonstrate that they have an ecological structure consistent with a recent living raft colony [34,46–48]. These extant rafts of marine debris deliver substantial communities of adult organisms capable of reproduction or colonization by zooplankton in marginal marine environments from one continental margin to another [5]. Rafts tend to be one-way arrival and deposition events which limit their journey time and lifespan before they encounter another system. These colonies are slow-moving (1–2 knots) compared with modern commercial vessels (20 to greater than 25 knots). These data are consistent with palaeoenvironmental interpretations suggesting an inhospitable seabed [49], indicating that these colonies remained afloat to survive and developed mostly in isolation and could easily have spread around the globe. Our data suggest that the life of the colony was far more dependent on the wood structure than previously suggested. Although the total weight of the crinoids would finally contribute to the sinking of the system, they are comparatively lightweight organisms compared to the bivalves.

# 5. Conclusion

In the early Mesozoic seas, these rafts were home to these large pseudoplanktonic communities. Their existence was a necessity as a result of a high number of anoxic shallow water basins that meant the benthic mode of life was not available to crinoids on the sea floor [49,50]. Therefore, the development of this lifestyle ensured the continued success of the group enabling the re-colonization into hospitable benthic environments following the Toarcian Oceanic Anoxic Event and their survival until today.

Data accessibility. Extended methods for the spatial analysis and the diffusion analysis as well as properties, definitions and equations are discussed in detail in the electronic supplementary material.

Authors' contributions. A.W.H., D.C. and E.G.M. contributed equally to research, discussion and manuscript preparation. A.W.H designed the research project, collected the museum data, photographed fossil material and prepared the specimen figures including the spatial analysis plots. D.C. designed and ran the diffusion analyses, and prepared the corresponding diagrams. E.G.M. designed and ran the spatial analysis, and prepared the corresponding diagrams. C.M. prepared the reconstruction.

Competing interests. The authors declare that they have no competing interests.

Funding. E.G.M. was funded by a Junior Research Fellowship from Murray Edwards College, the University of Cambridge, NERC standard grant NERC(NE/P002412/1) and NERC Independent research fellowship(NE/S014756/1) during the preparation of this manuscript. A.W.H. acknowledges DAAD funding in enabling the collection of data.

Acknowledgements. We thank the following institutions for access to the specimens: Geoscience Centre of the University of Göttingen; Werkforum Museum, Dotternhausen; Naturmuseum Senckenberg, Frankfurt am Main; Geological Institute, University of Tuebingen; Staatliches Museum für Naturkunde, Stuttgart; and Urweltmuseum Hauff, Holzmaden. We thank T. A. M. Ewin, Natural History Museum, London, for his help with the early stages of the project. Also special thanks to R. Haude, University of Göttingen, for providing the composite image of Holzmaden log (G1) (figure 1a; electronic supplementary material, figure S1). We thank the Associate Editors Prof. Stephen Hesselbo and Prof. Tatsuo Oji and an anonymous reviewer for their very useful comments and suggestions. We acknowledge support from DAAD funding enabling the collection of data (A.W.H.) and a Junior Research Fellowship from Murray Edwards College, University of Cambridge (E.G.M.).

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
