## [Reviewer comments · Royal Society Open Science]

Review History

RSOS-200142.R0 (Original submission)

Review form: Reviewer 1 (Tatsuo Oji)

Is the manuscript scientifically sound in its present form?

No

Are the interpretations and conclusions justified by the results?

No

Is the language acceptable?

Yes

Do you have any ethical concerns with this paper?

No

Have you any concerns about statistical analyses in this paper?

Yes

Recommendation?

Major revision is needed (please make suggestions in comments)

Comments to the Author(s)

Challenging study for the old but not-settled problem! As you know, I do not favor the pseudoplanktonic interpretation, and after reading your manuscript, unfortunately I was not convinced by your interpretation. I made some annotations in the text (Appendix A). The first "evidence" you provided, that attachment of crinoids onto the log is preferential to one side of the log, is also explained by half-buried log on the sea floor. I understand that the log could be float for a long time for the crinoids to grow to mature. Other questions are:

1. How the juvenile crinoids attach to the log? They attach as a larva and then extended their stalk? If so, the distalmost stalk near the attachment should be very slender, not being able to support the entire animal. Once the crinoids were detached from the log, they could not swim and re-attach the log to attach.
2. If the crinoids used a log for attachment, smaller log should have smaller individuals and larger log have larger individuals, because the larger log can be float for longer time. You mentioned a little about this aspect in the discussion but did not provided the data. Could you observe such a tendency?

I look forward to your updated manuscript with more convincing evidence and data.
Tatsuo Oji

Review form: Reviewer 2

Is the manuscript scientifically sound in its present form?

No

Are the interpretations and conclusions justified by the results?

No

Is the language acceptable?

Yes

Do you have any ethical concerns with this paper?

No

Have you any concerns about statistical analyses in this paper?

No

Recommendation?

Major revision is needed (please make suggestions in comments)

Comments to the Author(s)

This ms is a palaeoecological analysis of six fossil logs encrusted with the supposed pseudoplanktonic crinoid *Seirocrinus* from the Toarcian of Germany, using spatial analysis, together with modelling of time to saturation of wood, to estimate potential floating time for the studied logs. Whilst there have been a number of other studies on very similar material (e.g. Matzke and Maisch, 2019, listed in the references), the spatial analysis and wood saturation modelling is novel, and add to the discussion about the pseudoplanktonic habit of *Seirocrinus*. Overall the ms is understandable and fairly well illustrated, but I have some issues with some aspects of it that I think require further work from the authors.

Detailed comments:

Line 1. In the title and elsewhere in the text the authors use the term 'megaraft' colony. This is a misleading term for several reasons. First is that floating logs are not rafts, which are human

constructions. Second, the analysis in the paper is based on six log specimens, several of which are 'small' (line 51), so are these also 'mega'? In which case what defines a 'mega' raft? I suggest getting rid of this term and just calling these what they are: floating, encrusted logs of various sizes. Another related issue is the use of the singular 'megaraft colony' in the title, when the analysis is based on six logs.

Line 27. Here and elsewhere in the text the authors claim the encrusted floating logs are 'amongst the largest in-situ invertebrate accumulates ever to exist in the Phanerozoic fossil record'. I'm sure people who work on Phanerozoic reefs would be surprised to hear this, given many of these stretch 10s of kilometres. Because of this, I strongly suggest the authors remove this statement from the text. It's wrong and doesn't add anything to the paper.

Lines 28-29. What evidence is there for the Toarcian floating oceanic niche 'relatively predator free'? There are plenty of wonderfully preserved fish and Marine Mesozoic Reptile specimens (e.g. ichthyosaurs, plesiosaurs and crocodiles) in Toarcian deposits in Germany, as well as cephalopods, that presumably could have nibbled away pseudoplanktonic invertebrates on a floating log.

Line 32. It would be helpful for clarity for the reader to be told that the moisture diffuse models relate to how long wood can remain afloat.

Line 33. What is a 'migrating structure'? Presumably this means a floating log. In which case say so for clarity. Migrating brings to mind animal movement under their own power, which is not really the same as a log drifting around passively under the influence of waves and wind.

Line 47. Is the 6 year data for wood only? What about other floating objects like pumice and plastic?

Line 48. The statement that 'the deep time ecology of these communities has never been studied in detail' is at odds with the number of references later in the ms on exactly the same subject. Are the authors suggesting that works like Matzke and Maisch, 2019 are not detailed?

Line 49. Barnacles.

Line 51. Modern examples of what? Crinoids or pseudoplanktonic communities?

Line 60. See point about line 27.

Line 61. What's the evidence for 'transoceanic' rafting? I very much doubt this can be proven in the fossil record.

Line 64. Which hypothesis? The line above gives two alternatives.

Line 69. As per comment on line 32.

Lines 110-111. I feel the authors need do some more research on wood types present in the Toarcian European epicontinental seaways. This data exists in various publications (including Matzke and Maisch, 2019, page 98, and on the Bornholm sections in Denmark). Also, the growth rate for *Pseudomytloides dubius* (the byssate inoceramid referred to here) has been estimated by Caswell and Coe, 2013.

Line 116. The use of oysters to estimate the biomass of *Pseudomytloides dubius* is flawed in my view, as oysters produce substantially thicker shells than did the inoceramids. Therefore, the weight of an equivalent number of *Pseudomytloides dubius* specimens to oysters will be less (perhaps considerably so), and so the wood log to which they were attached would have remained floating for longer. A number of publications give the size ranges of *Pseudomytloides*

dubius (e.g. Caswell and Coe, 2013) and other publications give the thickness of inoceramid shells, so I think an estimation of *Pseudomytiloides dubius* biomass could be made.

Lines 133, 139, 140. See point above about the use of oysters as proxies for inoceramids.

Line 147. 'Left hand side'. This is confusing. Compared to what? Wood has no directionality. Presumably, the authors are referring to the relative position on the figure.

Line 155. A question here. How is it known that the small logs are complete specimens?

Line 168-169. It is very dubious to say that modern oysters are the 'nearest modern equivalent' to *Pseudomytiloides dubius* (see points above). The authors should use literature on the growth rates estimated for that species from the literature.

Lines 178-182. I fail to see how the decrease in density along the log of *Seirocrinus* specimens proves the colony was attached to the log while it was floating rather than on the seafloor (benthic). Surely differential density of specimens could have occurred on a sunken log too. This is not to say I question a pseudoplanktonic habit for *Seirocrinus*, it is just I don't think the spatial analysis proves the point.

Lines 183-186. I really don't understand the logic in these sentences. Wood logs float passively in the sea. They are not 'driven' like boats, so the analogy of a 'front' and 'back' to a log does not make sense. The 'drivers' for floating logs will be wind and waves, which act across the whole of log, depending for wind on how much of the log projects out of the water. Thus, a log is more likely to be pushed horizontally in the water column, so will not have a 'font' or 'back'.

Line 187. Could the authors expand on the evidence that the *Seirocrinus* specimens were attached onto the underside of the log.

Line 218. What the process by which *Seirocrinus* specimens seal wood, preventing water ingress? They did not have substantial holdfasts, being instead attached using cirri, so I don't see how they would have provided much in the way of sealing.

Lines 236-238. This goes back to the question from line 155. Another explanation for large specimens on small pieces of wood is that only a small piece of the available wood was collected, i.e. only the bit with the nice specimen. Wood from the Posidonienscheifer is difficult to collect because the surrounding shale tend to flake into tiny pieces.

Lines 249-250. These statements have been made before – repetition.

Line 262. The conclusion section should just be a precis of information and discussion already given in the rest of the paper. This conclusion contains additional thoughts that should be in the discussion, and is therefore not a conclusion.

Line 265. Given that the studied Toarcian logs had a maximum of two pseudoplanktonic species (*Seirocrinus subangularis* and *Pseudomytiloides dubius*) how can these 'rafts' be 'larger' and 'more complex' than modern floating wood communities, which can have many, many more taxa on them, on very large logs.

Line 266. How was the existence of the *Seirocrinus subangularis* floating colonies a 'necessity as a result of a high number of anoxic shallow water basins'? Is the point being made here that during the Toarcian OAE the benthic mode of life was not available to crinoids in these basins?

Lines 271-272. What has angiosperm wood got to do with this story, given angiosperms did not evolve until the early Cretaceous? Also, what is the unique ecological phenomenon being referred to? This can't be pseudoplankton, which are just a prevalent today as in the Toarcian.

Reference

Bryony A. Caswell and Angela L. Coe 2013. Primary productivity controls on opportunistic bivalves during Early Jurassic oceanic deoxygenation. *Geology*, doi:10.1130/G34819.1.

Decision letter (RSOS-200142.R0)

27-Mar-2020

Dear Dr Hunter,

The editors assigned to your paper ("Reconstructing the ecology of a Jurassic pseudoplanktonic megaraft colony") have now received comments from reviewers. We would like you to revise your paper in accordance with the referee and Associate Editor suggestions which can be found below (not including confidential reports to the Editor). Please note this decision does not guarantee eventual acceptance.

The reviews are generally supportive, but please note that your tendency to hyperbole in some of the statements detracts from the overall impact of the work. I would also urge you to consider that preservation potential, rather than raft availability, played a role in your interesting findings. It is also clear that some typographical errors and errors of omission need to be addressed.

Please submit a copy of your revised paper before 19-Apr-2020. Please note that the revision deadline will expire at 00.00am on this date. If we do not hear from you within this time then it will be assumed that the paper has been withdrawn. In exceptional circumstances, extensions may be possible if agreed with the Editorial Office in advance. We do not allow multiple rounds of revision so we urge you to make every effort to fully address all of the comments at this stage. If deemed necessary by the Editors, your manuscript will be sent back to one or more of the original reviewers for assessment. If the original reviewers are not available, we may invite new reviewers.

- Data accessibility

<http://datadryad.org/submit?journalID=RSOS&manu=RSOS-200142>

- Competing interests

- Authors' contributions

- Acknowledgements

- Funding statement

Kind regards,

Lianne Parkhouse
Royal Society Open Science
openscience@royalsociety.org

on behalf of Professor Stephen Hesselbo (Associate Editor) and Jon Blundy (Subject Editor)
openscience@royalsociety.org

Associate Editor's comments (Professor Stephen Hesselbo):

This manuscript quantitatively explores the processes that lead to sinking of communities that were hypothetically attached to floating logs and recovered from the Jurassic Holzmaden deposit, Germany. As highlighted by the referees, the study is an original contribution to debate on the nature of ancient pseudo-planktonic ecosystems, but there are some significant questions still to be answered. Additionally, both the main text and the supplementary text contain a significant number of typos and punctuation errors, inconsistent use of American and British English, as well as hyperbolic language (why 'massive giant' and not 'massive', or 'giant', or even just 'large'?). The supplementary text also requires figure captions that are more explanatory, and some of the supplementary material may usefully be transferred into the main text to support the arguments and to ensure consistency; in particular discussion of the wood structure and its behaviour during waterlogging, and details of the modelling assumptions are fundamental to the overall conclusions. Finally, the authors should consider that the existence of these large ecosystems in the Early Toarcian black shale environments is simply due to enhanced preservation potential, not because the environment was ecologically favourable to development of floating log communities.

Reviewers' Comments to Author:

Reviewer: 1

Comments to the Author(s)

Challenging study for the old but not-settled problem! As you know, I do not favor the pseudoplanktonic interpretation, and after reading your manuscript, unfortunately I was not convinced by your interpretation. I made some annotations in the text (attached).

The first "evidence" you provided, that attachment of crinoids onto the log is preferential to one side of the log, is also explained by half-buried log on the sea floor. I understand that the log could be float for a long time for the crinoids to grow to mature. Other questions are:

1. How the juvenile crinoids attach to the log? They attach as a larva and then extended their stalk? If so, the distalmost stalk near the attachment should be very slender, not being able to support the entire animal. Once the crinoids were detached from the log, they could not swim and re-attach the log to attach.
2. If the crinoids used a log for attachment, smaller log should have smaller individuals and larger log have larger individuals, because the larger log can be float for longer time. You mentioned a little about this aspect in the discussion but did not provided the data. Could you observe such a tendency?

I look forward to your updated manuscript with more convincing evidence and data.

Tatsuo Oji

Reviewer: 2

Comments to the Author(s)

This ms is a palaeoecological analysis of six fossil logs encrusted with the supposed pseudoplanktonic crinoid *Seirocrinus* from the Toarcian of Germany, using spatial analysis, together with modelling of time to saturation of wood, to estimate potential floating time for the studied logs. Whilst there have been a number of other studies on very similar material (e.g. Matzke and Maisch, 2019, listed in the references), the spatial analysis and wood saturation

modelling is novel, and add to the discussion about the pseudoplanktonic habit of *Seirocrinus*. Overall the ms is understandable and fairly well illustrated, but I have some issues with some aspects of it that I think require further work from the authors.

Detailed comments:

Line 1. In the title and elsewhere in the text the authors use the term 'megaraft' colony. This is a misleading term for several reasons. First is that floating logs are not rafts, which are human constructions. Second, the analysis in the paper is based on six log specimens, several of which are 'small' (line 51), so are these also 'mega'? In which case what defines a 'mega' raft? I suggest getting rid of this term and just calling these what they are: floating, encrusted logs of various sizes. Another related issue is the use of the singular 'megaraft colony' in the title, when the analysis is based on six logs.

Line 27. Here and elsewhere in the text the authors claim the encrusted floating logs are 'amongst the largest in-situ invertebrate accumulates ever to exist in the Phanerozoic fossil record'. I'm sure people who work on Phanerozoic reefs would be surprised to hear this, given many of these stretch 10s of kilometres. Because of this, I strongly suggest the authors remove this statement from the text. It's wrong and doesn't add anything to the paper.

Lines 28-29. What evidence is there for the Toarcian floating oceanic niche 'relatively predator free'? There are plenty of wonderfully preserved fish and Marine Mesozoic Reptile specimens (e.g. ichthyosaurs, plesiosaurs and crocodiles) in Toarcian deposits in Germany, as well as cephalopods, that presumably could have nibbled away pseudoplanktonic invertebrates on a floating log.

Line 32. It would be helpful for clarity for the reader to be told that the moisture diffuse models relate to how long wood can remain afloat.

Line 33. What is a 'migrating structure'? Presumably this means a floating log. In which case say so for clarity. Migrating brings to mind animal movement under their own power, which is not really the same as a log drifting around passively under the influence of waves and wind.

Line 47. Is the 6 year data for wood only? What about other floating objects like pumice and plastic?

Line 48. The statement that 'the deep time ecology of these communities has never been studied in detail' is at odds with the number of references later in the ms on exactly the same subject. Are the authors suggesting that works like Matzke and Maisch, 2019 are not detailed?

Line 49. Barnacles.

Line 51. Modern examples of what? Crinoids or pseudoplanktonic communities?

Line 60. See point about line 27.

Line 61. What's the evidence for 'transoceanic' rafting? I very much doubt this can be proven in the fossil record.

Line 64. Which hypothesis? The line above gives two alternatives.

Line 69. As per comment on line 32.

Lines 110-111. I feel the authors need do some more research on wood types present in the Toarcian European epicontinental seaways. This data exists in various publications (including Matzke and Maisch, 2019, page 98, and on the Bornholm sections in Denmark). Also, the growth

rate for *Pseudomytiloides dubius* (the byssate inoceramid referred to here) has been estimated by Caswell and Coe, 2013.

Line 116. The use of oysters to estimate the biomass of *Pseudomytiloides dubius* is flawed in my view, as oysters produce substantially thicker shells than did the inoceramids. Therefore, the weight of an equivalent number of *Pseudomytiloides dubius* specimens to oysters will be less (perhaps considerably so), and so the wood log to which they were attached would have remained floating for longer. A number of publications give the size ranges of *Pseudomytiloides dubius* (e.g. Caswell and Coe, 2013) and other publications give the thickness of inoceramid shells, so I think an estimation of *Pseudomytiloides dubius* biomass could be made.

Lines 133, 139, 140. See point above about the use of oysters as proxies for inoceramids.

Line 147. 'Left hand side'. This is confusing. Compared to what? Wood has no directionality. Presumably, the authors are referring to the relative position on the figure.

Line 155. A question here. How is it known that the small logs are complete specimens?

Line 168-169. It is very dubious to say that modern oysters are the 'nearest modern equivalent' to *Pseudomytiloides dubius* (see points above). The authors should use literature on the growth rates estimated for that species from the literature.

Lines 178-182. I fail to see how the decrease in density along the log of *Seirocrinus* specimens proves the colony was attached to the log while it was floating rather than on the seafloor (benthic). Surely differential density of specimens could have occurred on a sunken log too. This is not to say I question a pseudoplanktonic habit for *Seirocrinus*, it is just I don't think the spatial analysis proves the point.

Lines 183-186. I really don't understand the logic in these sentences. Wood logs float passively in the sea. They are not 'driven' like boats, so the analogy of a 'front' and 'back' to a log does not make sense. The 'drivers' for floating logs will be wind and waves, which act across the whole of log, depending for wind on how much of the log projects out of the water. Thus, a log is more likely to be pushed horizontally in the water column, so will not have a 'font' or 'back'.

Line 187. Could the authors expand on the evidence that the *Seirocrinus* specimens were attached onto the underside of the log.

Line 218. What the process by which *Seirocrinus* specimens seal wood, preventing water ingress? They did not have substantial holdfasts, being instead attached using cirri, so I don't see how they would have provided much in the way of sealing.

Lines 236-238. This goes back to the question from line 155. Another explanation for large specimens on small pieces of wood is that only a small piece of the available wood was collected, i.e. only the bit with the nice specimen. Wood from the Posidonienscheifer is difficult to collect because the surrounding shale tend to flake into tiny pieces.

Lines 249-250. These statements have been made before – repetition.

Line 262. The conclusion section should just be a precis of information and discussion already given in the rest of the paper. This conclusion contains additional thoughts that should be in the discussion, and is therefore not a conclusion.

Line 265. Given that the studied Toarcian logs had a maximum of two pseudoplanktonic species (*Seirocrinus subangularis* and *Pseudomytiloides dubius*) how can these 'rafts' be 'larger' and 'more complex' than modern floating wood communities, which can have many, many more taxa on them, on very large logs.

Line 266. How was the existence of the *Seirocrinus subangularis* floating colonies a ‘necessity as a result of a high number of anoxic shallow water basins’? Is the point being made here that during the Toarcian OAE the benthic mode of life was not available to crinoids in these basins?

Lines 271-272. What has angiosperm wood got to do with this story, given angiosperms did not evolve until the early Cretaceous? Also, what is the unique ecological phenomenon being referred to? This can’t be pseudoplankton, which are just a prevalent today as in the Toarcian.

Reference

Bryony A. Caswell and Angela L. Coe 2013. Primary productivity controls on opportunistic bivalves during Early Jurassic oceanic deoxygenation. *Geology*, doi:10.1130/G34819.1.

Author's Response to Decision Letter for (RSOS-200142.R0)

See Appendix B.

RSOS-200142.R1 (Revision)

Review form: Reviewer 1 (Tatsuo Oji)

Is the manuscript scientifically sound in its present form?

Yes

Are the interpretations and conclusions justified by the results?

No

Is the language acceptable?

Yes

Do you have any ethical concerns with this paper?

No

Have you any concerns about statistical analyses in this paper?

No

Recommendation?

Accept with minor revision (please list in comments)

Comments to the Author(s)

I found that this ms has been improved, but still there are some places where you may explain in more details, such as the followings.

Line

62: these rafts they represent one of the....  these rafts represent one of the...

63: [6], They...  [6]. They... (change comma to period)

67: Pseudoplanktonic Hypothesis  Pseudoplanktonic hypothesis

102-:

randomly distributed points can be modelled using homogeneous Poisson process and are found, for example, when larvae attach to....

 Initial settlement of larvae might be explained by Poisson process, but later there should be "struggle for survival" among individuals during growth because some are very closely positioned and could not survive successfully. Therefore, I recommend the authors to comment on this later process, not only talking about the initial settlement. Also the authors are advised to comment if all the crinoids on the log attached to the log at the same time, or gradually the log had gathered individuals at different times. This may change the scenario of the authors' spatial analysis.

220-238:

In the "Discussion", authors are advised to state clearly why benthic lifestyle cannot be explained by the spatial analysis. As I comment on the first version, half-buried log could host preferentially attached individuals on the log. Your bare surface on the upper side of the log might be the underside of the log where crinoids could not settle on.

[END]

Review form: Reviewer 2

Is the manuscript scientifically sound in its present form?

Yes

Are the interpretations and conclusions justified by the results?

Yes

Is the language acceptable?

Yes

Do you have any ethical concerns with this paper?

No

Have you any concerns about statistical analyses in this paper?

No

Recommendation?

Accept with minor revision (please list in comments)

Comments to the Author(s)

The authors have done a reasonable job amending the ms in relation to my review comments, and, in particular, I am pleased they used *Pseudomytiloides* data in their modelling. They have toned down the hyperbole although I still think the phrase 'largest in-situ monospecific/deuterostome invertebrate accumulations ever to exist in the Phanerozoic fossil record' is overwrought and unnecessary (as well as now being even more of a mouthful), as I can think of other similar accumulations, such as oyster reefs and serpulid reefs, both of which can be huge (100 m long and metres thick).

One issue that I still initially disagreed with the authors on was this comment on their first ms draft:

Lines 183-186. I really don't understand the logic in these sentences. Wood logs float passively in the sea. They are not 'driven' like boats, so the analogy of a 'front' and 'back' to a log does not

make sense. The 'drivers' for floating logs will be wind and waves, which act across the whole of log, depending for wind on how much of the log projects out of the water. Thus, a log is more likely to be pushed horizontally in the water column, so will not have a 'font' or 'back'.

To test this I did a small-scale experiment in my bathroom sink with a matchstick and a small piece of Blu-tack. First, I used just the matchstick, blowing it from one side of the sink to the other to see what orientation it moved in, alternating the starting orientation of the matchstick. The resulting movements were quite random, but, as I suspected, quite often the matchstick moved at right angles to my blowing (simulating wind over water). I then tried the same thing with my cupped hand simulating waves, with similar results. Then I added a small amount of Blu-tack to one end of the matchstick, enough to cause that end of the matchstick to sink slightly underwater and the unweighted end to raise slightly out of the water. This was to simulate the case in the paper of crinoids preferentially attached to one end of the log. When I then simulated wind and waves I found a very interesting thing, and that was regardless of the starting orientation, the matchstick always rotated and then moved with the weighted end trailing (the 'back') the unweighted end leading (the 'front'). I presume this is because the weighted end acts with greater friction in the water. This is exactly what is described in the ms, so I take back my initial criticism!

Decision letter (RSOS-200142.R1)

Dear Dr Hunter:

On behalf of the Editors, I am pleased to inform you that your Manuscript RSOS-200142.R1 entitled "Reconstructing the ecology of a Jurassic pseudoplanktonic megaraft colony" has been accepted for publication in Royal Society Open Science subject to minor revision in accordance with the referee suggestions. Please find the referees' comments at the end of this email.

The reviewers and Subject Editor have recommended publication, but also suggest some minor revisions to your manuscript. Therefore, I invite you to respond to the comments and revise your manuscript.

- Ethics statement

- Data accessibility

If you wish to submit your supporting data or code to Dryad (<http://datadryad.org/>), or modify your current submission to dryad, please use the following link:
<http://datadryad.org/submit?journalID=RSOS&manu=RSOS-200142.R1>

- **Competing interests**

- **Authors' contributions**

- **Acknowledgements**

- **Funding statement**

Because the schedule for publication is very tight, it is a condition of publication that you submit the revised version of your manuscript before 12-Jun-2020. Please note that the revision deadline will expire at 00.00am on this date. If you do not think you will be able to meet this date please let me know immediately.

Kind regards,

Anita Kristiansen
Editorial Coordinator

on behalf of Professor Stephen Hesselbo (Associate Editor)
openscience@royalsociety.org

Associate Editor Comments to Author (Professor Stephen Hesselbo):

Comments to the Author:

The manuscript has been significantly revised by the authors in line with reviewers' comments. There are now a couple of minor points still to address, which should be fairly easily accomplished.

Reviewer comments to Author:
Reviewer: 1

Comments to the Author(s)

I found that this ms has been improved, but still there are some places where you may explain in more details, such as the followings.

Line

62: these rafts they represent one of the....  these rafts represent one of the...

63: [6], They...  [6]. They... (change comma to period)

67: Pseudoplanktonic Hypothesis  Pseudoplanktonic hypothesis

102-:

randomly distributed points can be modelled using homogeneous Poisson process and are found, for example, when larvae attach to....

 Initial settlement of larvae might be explained by Poisson process, but later there should be "struggle for survival" among individuals during growth because some are very closely positioned and could not survive successfully. Therefore, I recommend the authors to comment on this later process, not only talking about the initial settlement. Also the authors are advised to comment if all the crinoids on the log attached to the log at the same time, or gradually the log had gathered individuals at different times. This may change the scenario of the authors' spatial analysis.

220-238:

In the "Discussion", authors are advised to state clearly why benthic lifestyle cannot be explained by the spatial analysis. As I comment on the first version, half-buried log could host preferentially attached individuals on the log. Your bare surface on the upper side of the log might be the underside of the log where crinoids could not settle on.

[END]

Reviewer: 2

Comments to the Author(s)

The authors have done a reasonable job amending the ms in relation to my review comments, and, in particular, I am pleased they used *Pseudomytiloides* data in their modelling. They have toned down the hyperbole although I still think the phrase 'largest in-situ monospecific/deuterostome invertebrate accumulations ever to exist in the Phanerozoic fossil record' is overwrought and unnecessary (as well as now being even more of a mouthful), as I can think of other similar accumulations, such as oyster reefs and serpulid reefs, both of which can be huge (100 m long and metres thick).

One issue that I still initially disagreed with the authors on was this comment on their first ms draft:

Lines 183-186. I really don't understand the logic in these sentences. Wood logs float passively in the sea. They are not 'driven' like boats, so the analogy of a 'front' and 'back' to a log does not make sense. The 'drivers' for floating logs will be wind and waves, which act across the whole of log, depending for wind on how much of the log projects out of the water. Thus, a log is more likely to be pushed horizontally in the water column, so will not have a 'font' or 'back'.

To test this I did a small-scale experiment in my bathroom sink with a matchstick and a small piece of Blu-tack. First, I used just the matchstick, blowing it from one side of the sink to the other to see what orientation it moved in, alternating the starting orientation of the matchstick. The resulting movements were quite random, but, as I suspected, quite often the matchstick moved at right angles to my blowing (simulating wind over water). I then tried the same thing with my cupped hand simulating waves, with similar results. Then I added a small amount of Blu-tack to one end of the matchstick, enough to cause that end of the matchstick to sink slightly underwater and the unweighted end to raise slightly out of the water. This was to simulate the case in the paper of crinoids preferentially attached to one end of the log. When I then simulated wind and waves I found a very interesting thing, and that was regardless of the starting orientation, the matchstick always rotated and then moved with the weighted end trailing (the 'back') the unweighted end leading (the 'front'). I presume this is because the weighted end acts with greater friction in the water. This is exactly what is described in the ms, so I take back my initial criticism!

Author's Response to Decision Letter for (RSOS-200142.R1)

See Appendix C.

Decision letter (RSOS-200142.R2)

Dear Dr Hunter,

It is a pleasure to accept your manuscript entitled "Reconstructing the ecology of a Jurassic pseudoplanktonic megaraft colony" in its current form for publication in Royal Society Open Science.

on behalf of Professor Stephen Hesselbo (Associate Editor)

Appendix A**ROYAL SOCIETY
OPEN SCIENCE****Reconstructing the ecology of a Jurassic pseudoplanktonic
megaraft colony**

Journal:	Royal Society Open Science
Manuscript ID	RSOS-200142
Article Type:	Research
Date Submitted by the Author:	23-Jan-2020
Complete List of Authors:	Hunter, Aaron; University of Cambridge, Department of Earth Sciences; University of Western Australia Faculty of Science, School of Earth Sciences; University of Western Australia Faculty of Science, School of Earth Sciences Casenove, David; Chiba University, Graduate School of Nursing Mitchell, Emily; University of Cambridge, Department of Zoology Mayers, Celia Mayers; Curtin University, Department of Applied Geology
Subject:	Palaeontology < EARTH SCIENCES, palaeontology < BIOLOGY, ecology < BIOLOGY
Keywords:	Transoceanic rafting, Holzmaden Lagerstätte, Spatial Analysis, Diffusion analyses, Crinoidea
Subject Category:	Earth and Environmental Science

Author-supplied statements

Relevant information will appear here if provided.

Ethics

Does your article include research that required ethical approval or permits?:

This article does not present research with ethical considerations

Statement (if applicable):

CUST_IF_YES_ETHICS :No data available.

Data

It is a condition of publication that data, code and materials supporting your paper are made publicly available. Does your paper present new data?:

Yes

Statement (if applicable):

Data is within ESM

Conflict of interest

I/We declare we have no competing interests

Statement (if applicable):

CUST_STATE_CONFLICT :No data available.

Authors' contributions

This paper has multiple authors and our individual contributions were as below

Statement (if applicable):

A.W.H., D.C. and E.G.M. contributed equally to research, discussion, and manuscript preparation.

A.W.H designed the research project, collected the museum data, photographed fossil material and prepared the specimen figures including the spatial analysis plots. D.C. designed and ran the diffusion analyses, and prepared the corresponding diagrams. E.G.M. designed and ran the spatial analysis, and prepared the corresponding diagrams. C.M. prepared the reconstruction.

**Reconstructing the ecology of a Jurassic pseudoplanktonic megaraft**
**colony**

**Short title:** A Jurassic megaraft ecosystem

**Aaron W. Hunter^{1,2}, David Casenove³, Emily G. Mitchell¹ and Celia Mayers⁴**

¹Department of Earth Sciences, University of Cambridge, Downing Street, Cambridge, CB2 3EQ, UK.

²School of Earth Sciences, University of Western Australia, 35 Stirling Highway, Crawley, WA 6009, Australia.

³Graduate School of Nursing, Chiba University, 1-8-1, Inohana, Chuo-ku, Chiba-shi, Chiba, 260-8672, Japan

⁴Department of Applied Geology, Curtin University, Perth, Western Australia

**ORCID**

AWH, 0000-0002-3527-5835

**Author for correspondence:**

Aaron W. Hunter

e-mail:awh31@cam.ac.uk
24

Abstract

Pseudoplanktonic crinoid megaraft colonies are an enigma of the Jurassic. They are among the largest in-situ invertebrate accumulations ever to exist in the Phanerozoic fossil record. These megaraft colonies ~~and~~ are thought to have developed as floating filter-feeding communities due to an exceptionally rich relatively predator free oceanic niche, high in the water column enabling them to reach high densities on these log rafts. However, this pseudoplanktonic hypothesis has never actually been quantitatively tested and some researchers have cast doubt that this mode of life was even possible. The ecological structure of the crinoid colony is resolved using spatial point process analyses and its longevity using moisture diffusion models. Using spatial analysis we found that the crinoids would have trailed preferentially positioned at the back of migrating structures in the regions of least resistance, consistent with a floating, not benthic ecology. Additionally, we found using a series of moisture diffusion models at different log densities and sizes that ecosystem collapse did not take place solely due to colonies becoming overladen as previously assumed. We have found that these crinoid colonies studied could have existed for greater than 10 years, even up to 20 years exceeding the life expectancy of modern documented megaraft systems with implications for the role of modern raft communities in the biotic colonisation of oceanic islands and intercontinental dispersal of marine and terrestrial species.

Keywords

Transoceanic rafting | Holzmaden Lagerstätte | Spatial Analysis | Diffusion analyses | Crinoidea

1. Introduction

Transoceanic rafting is a fundamental feature of marine evolutionary biogeography and ecology, often invoked to explain the origins of modern global patterns of species distributions [1,2,3,4]. These communities have been recorded today lasting up to 6 years [5]. However, the deep time ecology of these communities has never been investigated in detail [6]. In recent communities, such rafts have

1 49 included highly adapted bivalves, barnacle, limpets, bryozoans, sea anemones, amphipods, and isopods
[5]. In the Jurassic these communities also consisted of specially adapted crinoids, whose apparent
maturity suggests that these communities had to have lasted longer than modern examples (>6 years)
[6]. The structure and duration and these colonies has remained unresolved, with most studies choosing
to focus on how the crinoids were adapted rather than the viability of the system, prompting debate on
their lifestyle [7,8] rather than the ecological structure and longevity of the habitat. This study uses the
latest ecological techniques used in paleobiology to reconstruct the ecology and duration of these rafts.

Crinoids or sea lilies were a major part of the Jurassic shallow sea ecosystem, with crinoids found
in a diverse suite of shallow marine environments [9]. The monospecific crinoid colonies preserved on
wood rafts are one of the most enigmatic and iconic of these communities [10]. They existed during the
Toarcian Oceanic Anoxic Event, a short-lived period of widespread dysoxic conditions [11]. Found
globally, they represent one of the largest *in-situ* invertebrate accumulations found in the fossil record
[6], the only fossil example of transoceanic rafting with up to 100 individuals covering
bivalve-encrusted logs up to 14m long [12]. It has been debated whether these crinoids could have
colonized and persisted on these floating log habitats or they were instead part of benthic islands
systems typical of the Mesozoic [13]. Recently this hypothesis has become widely accepted mainly
because (i) these crinoids are always preserved associated with large drift wood, (ii) they are found in
anoxic sediments, (iii) they possess morphological features critical for a pseudoplanktonic lifestyle
(long stem enabling efficient tow-net filtration). Although previous studies have examined the
distribution of crinoids on the logs [14,15] their distributions were not quantitatively described. In the
present study we use spatial statistics and diffusion modelling in a novel approach to test whether this
pseudoplanktonic mode of life existed. Spatial statistics are used to test whether the logs were colonized
in open water or on the substrate, and diffusion models then quantify floating-log mechanics to test how
long a floating system could have existed.

The spatial positions of *Seirocrinus* on one of the largest and best-preserved Early Jurassic floating
wood examples known, the giant ‘Hauff Specimen’ from Holzmaden, Germany [15] were mapped
(figure 1; electronic supplementary material, figures S1–S3). The spatial patterns of benthic organisms
depend on the dispersal of larvae [16,17], the environmental conditions in which they settled [18], and
whether the conditions were favorable for them to grow to adulthood [19,20]. Therefore, the spatial
patterns of the crinoids can be used to try to “reverse engineer” the conditions in which they settled in
order to deduce the environmental conditions (what part of the water column the log was in) when
colonized [21,22].

This analysis was complemented by analyses using density/diffusion models [23,24] to constrain
the length of time logs of differing sizes and densities, with a full colony, could stay afloat as water
infiltrates the wood over time. This diffusion analysis used three sets of size defined colonies (electronic
supplementary material, figures S4–S9), including small colony specimens (S1 and S2) (electronic
supplementary material, figures S4 and S5) medium colonies (M1 and M2) (electronic supplementary
material, figures S6 and S7) and finally massive giant colonies from Holzmaden (G1 “The Hauff
Specimen” and G2) (electronic supplementary material, figures S8 and S9).

43 89 2. Materials and methods

45 90 (a) Data collection

Photographs were taken from six specimens and composite images produced. These include small,
medium and finally massive giant colonies from the following German museums and collections:
Geoscience Centre of the University of Göttingen; Werkforum Museum, Dotternhausen; Naturmuseum
Senckenberg in Frankfurt am Main; Geological Institute, University of Tübingen; Staatliches Museum
für Naturkunde, Stuttgart; and Urweltmuseum Hauff, Holzmaden.

(b) Spatial analysis

Randomly distributed points (figure 1a) can be modelled using homogeneous Poisson processes, and are found, for example, when larvae attach to an substrate not impacted by strong currents nor patchy environmental variables [25]. While spatial processes that depend on position on the substrate can be modelled using heterogeneous Poisson processes, with density changing according to a given formula. To assess how disc density changed along the log, disc density was modelled as a heterogeneous Poisson process dependent on the x co-ordinate and then the y -co-ordinate. Model fit was assessed using the model residuals by plotting Q–Q and smoothed residual plots. If the observed line in the Q–Q plot fell outside two standard deviations of the model, the model was rejected [21,26]. Akaike information criterion values [27] were used to compare the relative quality of the statistical models that fitted the data.

(c) Moisture diffusion models

As it is not possible to know the actual properties of the fossil wood (due to the lack of fossil examples available for study) or the actual growth rate of the byssate inoceramids (as these species are now extinct), two types of models were developed. The first is a set of ‘catastrophic’ models (electronic supplementary material, figures S4–S9) which look at how long a series of size defined structures can survive with no natural sealing (see *Diffusion analyses* and electronic supplementary material, SI text). The second model is predictive, and proposes how long the mega-raft could have lasted if the system was efficiently sealed and a population of bivalves (based on extant oysters) was growing on its surface (see *Oyster community growth model* and electronic supplementary material, SI text). Both these models test the microstructure and density of the wood. For example dense woods will take longer to absorb water than less dense ones, and some wood types have transverse struts in the vessels called tyloses which slow waterlogging. The model is not able to test the osmotic pressure from seawater that

aids buoyancy or a process called bacterial sealing, whereby bacterial slime clogs vessels within the
wood [19]. These processes will increase the viability of the system and the length of time the log
colony would have lasted.

*Diffusion analyses.* For the first model, three sets of size defined colonies (electronic supplementary
material, figures S4–S9), including small colony specimens (S1 and S2) (electronic supplementary
material, figures S4 and S5) medium colonies (M1 and M2) (electronic supplementary material, figures
S6 and S7) and finally massive giant colonies from Holzmaden (G1 “The Hauff Specimen” and G2)
(electronic supplementary material, figures S8 and S9) were tested. To address the stability of the log
and colony, the density/diffusion model is constructed for each of the colony three-size classes using the
log properties without any maximum growth population attached, then the crinoid community is added,
along with a layer of bivalves (in this case extant oysters are used [12]). The models use a colony
weight based on modern isocrinid *Metacrinus* and Japanese oysters (*Crassostrea gigas*). They assume
there is no natural sealing of the wood and it begins to absorb water, and therefore breakdown, from the
onset of the colony (see electronic supplementary material, SI text).

*Oyster community growth model.* The second model considers the growth of the oyster community on
the surface of the same logs and estimates how long the log can remain afloat while bearing the weight
of the colony. It relies on oyster population data about extant communities of *Ostrea chilensis* and
*Crassostrea virginica*. The model of population growth assumes the settlement on one single individual
when the log contacts sea water and estimates the added mass of the community to the completely
sealed log every 1-year increment until the whole log and colony sinks due to its density (see electronic
supplementary material, figure S2 and Table S3).

3. Results

The spatial analyses clearly show an ecological signal of directionality of the positions of crinoid attachments along the log with the highest density on the left hand side, with decreasing density along the log (figure 1b; electronic supplementary material, figures S2 and S3). This directionality can be modelled by heterogeneous Poisson model of a linear model using depending on the x co-ordinate with AIC model comparisons resolving this model over other, more complex ones (electronic supplementary material, Table S1). The diffusion analysis is represented by two models, which constantly show that the largest of the log systems could have survived for a minimum of 2 years and a maximum of 20 years (figure 2; electronic supplementary material, Table S1) thus allowing the crinoid colony to grow to maturity and confirming the viability of the pseudoplanktonic hypothesis. First, the diffusion model assumes that the system is not adequately sealed. When the diffusion model is applied to the small logs (S1 and S2) (electronic supplementary material, figures S4 and S5) without the population, the log would sink after 800 days, however when the colony is added it would sink within 1 day (electronic supplementary material, figure S5). The apparent longevity of the logs is increased by their size, with the medium sized logs (M1 and M2) (electronic supplementary material, figures S6 and S7) surviving 100 days (800 without the colony) and the massive colonies (G1 and G2) (electronic supplementary material, figures S8 and S9) surviving up to 400 days (800 days without the colony) (electronic supplementary material, figure S9). Our results suggest that without any natural sealing the largest megarafts could only have survived for just over two years with the smaller systems unviable despite being relatively common. We propose that in order for the system to survive for long enough for the animals to grow to maturity the natural sealing of the wood structure would be needed with the bivalves and the crinoids being part of efficient sealing of the system. The population model incorporated both the spatial distribution along the log as well as life-history estimates (fecundity, mortality, settling rates and maturation time) for a complete life cycle using the extant oyster *Ostrea chilensis*, the nearest

modern equivalent to the byssate inoceramids found at Holzmaden. Simulations for the growth of oyster
communities including spawning rate, per capita reproductive rate and mortality rate correspond to
average values taken from the literature about *Ostrea chilensis*. The model allowed reconstruction of the
number of years required to reach sinking density for the whole log for various values of K (electronic
supplementary material, Tables S2 and S3). In some cases, the dimensions of the log combined with the
density of the wood material would allow the bivalves and crinoid loaded log never to reach sinking
threshold within the 20-year window (figure 2).

4. Discussion

The crinoids attached preferentially to one end of the log structure (figure 1*b*; electronic supplementary
material, figures S2 and S3), with a decrease of density down the log, and attachment discs barely
developed on the upper surface (figure 3). This result confirms an observation was also made in an
exceptionally preserved example from Holzmaden [8] and supports the hypothesis that our largest
colony was structured as a floating body not a benthic structure. This feature is demonstrated by modern
shipworms, for example, which preferentially accumulate at one end of the structure [28]. In the case of
a ship, this would be the back or stern. If the log were moving through the water even at a low rate the
bow of the structure would still be subject to the most current pressure, so the most hospitable part of
the structure would be the area of least resistance, the stern. However, this asymmetric distribution,
combined with the observation that the vast majority of crinoids are attached to the underside of the log
(figures 1*b* and 3; electronic supplementary material, figures S2 and S3), would make the log
unbalanced towards one end, bringing the colony's stability into question. The conventional view is that
the community would have rapidly sunk and been preserved at anoxic depths due to becoming unstable
or overladen [15]. However, our diffusion model does not support rapid sinking for three key reasons.

First, the growing community of bivalves and crinoids even at the climax state would still be a

193 minor component of the total weight of the system. This total maximum weight (in very rare cases)
would have varied from 630 to 880 kg depending on carry capacity and density compared to estimates
of between 1100 kg to 15,000 kg for the total mass of the log depending on wood type (electronic
supplementary material, SI text). Secondly, without efficient sealing our catastrophic models suggest
the log could have supported the population of crinoids for a substantial amount of time. Even with a
system designed to be a catastrophic scenario taking on water at a constant rate until the log raft system
failed, the largest examples (G1 and G2) (electronic supplementary material, figures S8 and S9) could
have survived for up to two years (electronic supplementary material, Table S3). Preserved fossil
examples in this study were close to their maximum point viability as a mobile substrate. This point is
evident from smaller logs (S1 and S2) (electronic supplementary material, figures S4 and S5) whose
population was simply too big to be supported by the smaller log structure. However, the model looks at
the viability of the climax community that would take time to develop, and would expand the life of the
colony within the parameters of the model >2 years. However, modern observed growth rates for recent
isocrinid crinoids 30-40 cm [29] in length suggests it would have taken at least 10 years for *Seirocrinus*
to reach maturity. Although the growth rates of this crinoid may have been faster due to the unusual
structure of the stem [30]. For these communities to exist to maturity the log needs to float for much
longer than 2 years (electronic supplementary material, figures S8 and S9). Our oyster community
growth model in contrast, predicts that the megaraft was viable for up to and exceeding 20 years (figure
2; electronic supplementary material, Table S3). This length of time would require the structure to be
sealed by a bivalve coating to preserve any failures in the wood structure. The key to understanding
how this lifespan of the megaraft was possible, is firstly by identifying areas of weaknesses where the
structure is most likely to fail, this would include areas on the surface not coated in bark and secondly
the ends of the log system that expose the wood xylem [31]. However, it is clear from all the examples
examined that most of the surface is covered in bivalves (figure 1) and as our results suggest that the

217 crinoids themselves are preferentially distributed at one terminal end of the system (figures 1*b* and 3;
electronic supplementary material, figures S2 and S3), contributing to the natural sealing of that end of
the log system. Not all areas of the log were sealed and this would have shortened the lifespan of the
community.

12221 Thirdly, the viability of the community would have been influenced by the natural properties of the
14222 wood itself. Although it is hard to model the wood structure with no fossil remains available in the
17223 Posidonia Shale [32], the gymnosperm wood structure with the addition of the bivalve coating and other
20224 agents such as aquatic fungus and algal slime must have allowed for sealing of the log and increased its
22225 longevity substantially beyond our models of continuous infiltration [33]. Estimates vary for how long
25226 modern gymnosperm pine driftwood can survive (see electronic supplementary material for full
27227 discussion). Some estimates are as little as 17 months [34] while others suggest that the wood could
30228 have remained buoyant up to 5 years after entering the marine environment (see electronic
33229 supplementary material). The exceptional properties of the Jurassic gymnosperm wood with natural
35230 sealing by the bivalves and crinoids, and without modern aggressive wood boring predators, which
38231 evolved later in the Mesozoic [35] the wood would have likely floated for longer than the 5 years
40232 observed in today's driftwood. The colony could therefore have survived for the 10 years needed for a
43233 community to reach maturity, and possibly longer, until the decay and water infiltration of the wood
46234 reached a limit that could no longer support any further community growth. At which point the log sank
48235 and the crinoids died, being perfectly preserved by the anoxic conditions on the seabed [12] (figure
51236 1*a,c*). Evidence suggests that fragments might have broken off while the system was in a state of decay
53237 which might explain the occurrence of single mature individuals attached to relatively small fragments
56238 of wood observed across the museum collections examined.

58
59239 Pseudoplanktonic organisms are highly significant for global ecology and biodiversity, as a
60
mechanism for global colonization [36], typically attaching themselves to nektonic and planktonic

organisms or other floating objects such as driftwood or flotsam. These are significant in today's
ecosystems and were responsible for the colonisation of oceanic islands such as Hawaii [37]. These
temporary rafts can become a permanent home to self-contained long-term communities such as, in
modern ecosystems, external goose barnacles, tunicates and bryozoans [38], with wood boring bivalves
such as shipworms inhabiting the internal structures [39].

Our analysis suggests that the exceptionally preserved Jurassic crinoid megarrafts are the longest
surviving pelagic rafting communities to exist in the fossil record. Our results not only demonstrate that
they have an ecological structure consistent with a recent living megarraft colony [28,40,41,42] (see
electronic supplementary material) but with efficient sealing these communities could have survived for
up to and exceeding 20 years. In contrast, those recorded today have survived for up to 6 years. These
extant megarrafts of marine debris deliver substantial communities of adult organisms capable of
reproduction or colonization by zooplankton in marginal marine environments from one continental
margin to another [5]. Rafts tend to be one-way arrival and deposition events which limit their journey
time and life span before they encounter another system. These colonies are slow moving (1 to 2 knots)
compared with modern commercial vessels (20 to >25 knots). These data are consistent with
palaeoenvironmental interpretations suggesting an inhospitable seabed [43], indicating that these
colonies remained afloat to survive and developed largely in isolation and could easily have spread
around the globe. Our data suggests that the life of the colony was far more dependent on the wood
structure than previously thought. Although the total weight of the crinoids would finally contribute to
sinking of the system, they are comparatively lightweight organisms compared to the bivalves.

53 261 54 55 56 262 **5. Conclusions**

57
58
59 263 We have demonstrated that Jurassic crinoids did inhabit this unique pseudoplanktonic niche, becoming
60
highly adapted, fast growing, very lightweight and self-sufficient viable communities. In the early

Mesozoic seas, these rafts were home to far larger and more complex, now lost, ecosystems whose
existence was a necessity as a result of a high number of anoxic shallow water basins [43,44].
Development of this lifestyle ensured the continued success of the group. The restoration of healthy
benthic environments following the Toarcian Oceanic Anoxic Event, followed by the appearance of
wood boring bivalves (9, 13). Although wood boring bivalves had appeared in the Middle Jurassic, they
only colonised wood already on the sea floor [36]. It was not until the evolution of teredinids in the
mid-late Cretaceous that bivalves first attacked driftwood [45]. Along with the prevalence of
angiosperm wood this unique ecological phenomenon vanished from the fossil record forever.

**Data accessibility**

Extended methods for the spatial analysis and the diffusion analysis as well as properties, definitions
and equations are discussed in detail in electronic supplementary material.

**Author' contributions**

281 A.W.H., D.C. and E.G.M. contributed equally to research, discussion, and manuscript preparation.
282 A.W.H designed the research project, collected the museum data, photographed fossil material and
283 prepared the specimen figures including the spatial analysis plots. D.C. designed and ran the diffusion
analyses, and prepared the corresponding diagrams. E.G.M. designed and ran the spatial analysis, and
prepared the corresponding diagrams. C.M. prepared the reconstruction.

**Funding**

E.G.M. was funded by a Junior Research Fellowship from Murray Edwards College, University of
Cambridge during the preparation of this manuscript. A.W.H. acknowledges DAAD funding in
enabling the collection of data.

11 12 13 292 **Competing Interests**

The authors declare that they have no competing interests.

22 23 295 **Acknowledgements**

We thank the following institutions for access to the specimens: Geoscience Centre of the University of
Göttingen; Werkforum Museum, Dotternhausen; Naturmuseum Senckenberg, Frankfurt am Main;
Geological Institute, University of Tuebingen; Staatliches Museum für Naturkunde, Stuttgart; and
Urweltmuseum Hauff, Holzmaden. We thank T Ewin, Natural History Museum, London, for his help
with the early stages of the project. Also special thanks to R Haude, University of Göttingen, for
providing the composite image of Holzmaden Log (G1) (figure 1*a*; electronic supplementary material,
figure S1). We acknowledge support from DAAD funding enabling collection of data (A.W.H.) and a
Junior Research Fellowship from Murray Edwards College, University of Cambridge (E.G.M.).

**Supplementary Material**

Electronic supplementary material includes supplementary text (SI text), nine supplementary figures
(S1–S9) and three tables (S1–S3).

**References**

[revised manuscript text omitted]

 45. Evans S. 1999 Wood-boring bivalves and boring linings. *B. Geol. Soc. Denmark* **45**, 130–134.

 **Figure captions**

 **Figure 1.** Crinoid fossil megaraft, the ‘Hauff Specimen’ from Holzmaden (G1). (a) Log with spatial
 analysis data points; key: blue= crinoid crowns, black= attachment discs. (b) Spatial analysis plot. (c)
 Close up view of crinoid crown and stem sections. (d) PCF distance plot.

 **Figure 2.** Wood diffusion gradient plots. (a) S1 unloaded. (b) S1 loaded. (c) M1 unloaded. (d) M1
 loaded. (e) G1 unloaded. (f) G1 loaded.

 **Figure 3.** Reconstruction of the Holzmaden ‘Hauff Specimen’ crinoid megaraft colony (G1).

Figure 1. Crinoid fossil megaraft, the 'Hauff Specimen' from Holzmaden (G1). (a) Log with spatial analysis data points; key: blue= crinoid crowns, black= attachment discs. (b) Spatial analysis plot. (c) Close up view of crinoid crown and stem sections. (d) PCF distance plot.

Figure 2. Wood diffusion gradient plots. (a) S1 unloaded. (b) S1 loaded. (c) M1 unloaded. (d) M1 loaded. (e) G1 unloaded. (f) G1 loaded.

Figure 3. Reconstruction of the Holzmaden 'Hauff Specimen' crinoid megaraft colony (G1).

Appendix B

19th April 2020

Editorial staff, *Royal Society Open Science*

Dear Editor

On behalf of myself and the authors, please find attached our revised manuscript entitled *Reconstructing the ecology of a Jurassic pseudoplanktonic megaraft colony*, which we hope you may consider for publication as a research article in *Royal Society Open Science*. We hope we have addressed all the comments and concerns from the Associate editor and the two reviewers and we have attached a reply to reviewers (also below) as well as a track changes version of the MS.

Thank you in advance for your consideration of our manuscript and best wishes.

Best wishes,

Aaron W. Hunter

On Behalf of David Casenove, Emily G. Mitchell and Celia Mayers

Associate Editor's comments (Professor Stephen Hesselbo):

This manuscript quantitatively explores the processes that lead to sinking of communities that were hypothetically attached to floating logs and recovered from the Jurassic Holzmaden deposit, Germany. As highlighted by the referees, the study is an original contribution to debate on the nature of ancient

pseudo-planktonic ecosystems, but there are some significant questions still to be answered. Additionally, both the main text and the supplementary text contain a significant number of typos and punctuation errors, inconsistent use of American and British English, as well as hyperbolic language (why ‘massive giant’ and not ‘massive’, or ‘giant’, or even just ‘large’?). The supplementary text also requires figure captions that are more explanatory, and some of the supplementary material may usefully be transferred into the main text to support the arguments and to ensure consistency; in particular discussion of the wood structure and its behaviour during waterlogging, and details of the modelling assumptions are fundamental to the overall conclusions. Finally, the authors should consider that the existence of these large ecosystems in the Early Toarcian black shale environments is simply due to enhanced preservation potential, not because the environment was ecologically favourable to development of floating log communities.

Thank you for these comments. We apologize for the errors, which are now rectified and have toned down the hyperbolic language. We have also expanded the figure captions and added a new figure. We have expanded our discussion on the model assumptions. The fossils in this study are from the Early-mid Toarcian *Posidonia Shale*. The Pseudoplanktonic model of life along with this style of preservation would have existed for *Seirocrinus* Pliensbachian-Toarcian and for *Pentacrinites* Hettangian-Bathonian. As such, our study is not influenced by the enhanced preservation as by the Bathonian these crinoids are preserved in shallow water carbonates. Unfortunately, the mode of life ceased to exist for these crinoids by the Middle Jurassic

Reviewers' Comments to Author:

Reviewer: 1

Comments to the Author(s)

Challenging study for the old but not-settled problem! As you know, I do not favor the pseudoplanktonic interpretation, and after reading your manuscript, unfortunately I was not convinced by your interpretation. I made some annotations in the text (attached).

Thank you for your comments. We have addressed the concerns of reviewer 1 and with these changes we believe we demonstrate that our MS provides evidence that this mode of life was viable.

The first "evidence" you provided, that attachment of crinoids onto the log is preferential to one side of the log, is also explained by half-buried log on the sea floor. I understand that the log could be float for a long time for the crinoids to grow to mature.

The attachment to the crinoids is not just on one side, but throughout. The spatial analyses show that it is significantly more dense on one side - the surface continuously inhabited by crinoids and the empty surface has attachment discs that did not develop but demonstrate that all the surfaces are exposed.

Other questions are:

1. How the juvenile crinoids attach to the log? They attach as a larva and then extended their stalk? If so, the distalmost stalk near the attachment should be very slender, not being able to support the entire animal. Once the crinoids were detached from the log, they could not swim and re-attach the log to attach.

We agree that this would work for modern Isocrinids. However these crinoids have a unique stem structure. They have a large attachment disk, and modified stalk structure. Thus they have an atypical growth rate and structure. There is no evidence that the stem is shed in these crinoids.

2. If the crinoids used a log for attachment, smaller log should have smaller individuals and larger log have larger individuals, because the larger log can be float for longer time. You mentioned a little about this aspect in the discussion but did not provided the data. Could you observe such a tendency?

That is a very interesting point, thank you. We will add this data to the MS.

I look forward to your updated manuscript with more convincing evidence and data.

Tatsuo Oji

Reviewer: 2

Comments to the Author(s)

This ms is a palaeoecological analysis of six fossil logs encrusted with the supposed pseudoplanktonic crinoid *Seirocrinus* from the Toarcian of Germany, using spatial analysis, together with modelling of time to saturation of wood, to estimate potential floating time for the studied logs. Whilst there have been a number of other studies on very similar material (e.g. Matzke and Maisch, 2019, listed in the references), the spatial analysis and wood saturation modelling is novel, and add to the discussion about the pseudoplanktonic habit of *Seirocrinus*. Overall the ms is understandable and fairly well illustrated, but I have some issues with some aspects of it that I think require further work from the authors.

Detailed comments:

Line 1. In the title and elsewhere in the text the authors use the term ‘megaraft’ colony. This is a misleading term for several reasons. First is that floating logs are not rafts, which are human constructions. Second, the analysis in the paper is based on six log specimens, several of which are ‘small’ (line 51), so are these also ‘mega’? In which case what defines a ‘mega’ raft? I suggest getting rid of this term and just calling these what they are: floating, encrusted logs of various sizes. Another related issue is the use of the singular ‘megaraft colony’ in the title, when the analysis is based on six logs.

That is a good point, we shall change to raft throughout which describes any type of floating body e.g mussel rafts, palm rafts.

Line 27. Here and elsewhere in the text the authors claim the encrusted floating logs are ‘amongst the largest in-situ invertebrate accumulates ever to exist in the Phanerozoic fossil record’. I’m sure people who work on Phanerozoic reefs would be surprised to hear this, given many of these stretch 10s of kilometres. Because of this, I strongly suggest the authors remove this statement from the text. It’s wrong and doesn’t add anything to the paper.

We are added the word monospecific/deuterostome as Phanerozoic reefs would be multi-taxon.

Lines 28-29. What evidence is there for the Toarcian floating oceanic niche ‘relatively predator free’? There are plenty of wonderfully preserved fish and Marine Mesozoic Reptile specimens (e.g. ichthyosaurs, plesiosaurs and crocodiles) in Toarcian deposits in Germany, as well as cephalopods, that presumably could have nibbled away pseudoplanktonic invertebrates on a floating log.

We agree and think it is best that we remove this unsupported statement.

Line 32. It would be helpful for clarity for the reader to be told that the moisture diffuse models relate to how long wood can remain afloat.

Corrected Line 33: estimates of the duration of the floating until sinking.

Line 33. What is a ‘migrating structure’? Presumably this means a floating log. In which case say so for clarity. Migrating brings to mind animal movement under their own power, which is not really the same as a log drifting around passively under the influence of waves and wind.

Corrected on lines 34.

Line 47. Is the 6 year data for wood only? What about other floating objects like pumice and plastic?

We only dealt with wood since this was the only object under considerations since there is no evidence of crinoids attached to other raft materials such as pumice.

Line 48. The statement that ‘the deep time ecology of these communities has never been studied in detail’ is at odds with the number of references later in the MS on exactly the same subject. Are the authors suggesting that works like Matzke and Maisch, 2019 are not detailed?

We have modified this as follows "*using the latest quantitative methods to empirically test these different hypotheses*".

Line 49. Barnacles.

Add Plural (S)

Line 51. Modern examples of what? Crinoids or pseudoplanktonic communities?

Added on lines XXX "*pseudoplanktonic communities*"

Line 60. See point about line 27.

See comment for Line 27 "*monospecific*" added

Line 61. What’s the evidence for ‘transoceanic’ rafting? I very much doubt this can be proven in the fossil record.

While it is hard to definitively prove anything in the fossil record, there are recent examples:

Carlton JT, Chapman JW, Geller JB, Miller JA, Carlton DA, McCuller MI, Treneman NC, Steves BP, Ruiz GM. 2017 Tsunami-driven rafting: Transoceanic species dispersal and implications for

marine biogeography. *Science* 357, 1402-1406.

As well as are published accounts of globally distributed pseudoplanktonic crinoids in the Jurassic.

Hunter AW, Oji T, & Okazaki Y (2011) The occurrence of the pseudoplanktonic crinoids Pentacrinites and Seirocrinus from the Early Jurassic Toyora Group, western Japan. *Paleontological research* 15(1):12–22.

Hunter AW & Zonneveld J-P (2008) Palaeoecology of Jurassic encrinites: reconstructing crinoid communities from the Western Interior Seaway of North America. *Palaeogeography, Palaeoclimatology, Palaeoecology* 263(1):58–70.

Line 64. Which hypothesis? The line above gives two alternatives.

We agree this is not clear we have stated “*Pseudoplanktonic Hypothesis*”.

Line 69. As per comment on line 32.

Model is related to how long these can float for.

Lines 110-111. I feel the authors need do some more research on wood types present in the Toarcian European epicontinental seaways. This data exists in various publications (including Matzke and Maisch, 2019, page 98, and on the Bornholm sections in Denmark). Also, the growth rate for *Pseudomytloides dubius* (the byssate inoceramid referred to here) has been estimated by Caswell and Coe, 2013.

Thank you for this point. Unfortunately wood types are not preserved, hence model needed.

In Matzke and Maisch (2019, p.98), the authors suggest three categories for the classification of fossil driftwood depending on levels of encrustation by bivalves and/or crinoids. Thin sections have been possible for only the non-encrusted wood logs (type A), allowing their classification as conifer “*Araucarioxylon*”. However, no classification has been possible for the other log types due to wood structures not being observable.

A more general examination of Early Jurassic vegetation in the Danish Basin suggests tree ferns (*Deltoidospora*) and pinacean conifers (*Pinuspollenites minimus*), along with common taxodiacean/cupressacean conifers, ginkgos/cycads and corystospermous seed ferns. See: Lindström, S., Erlström, M., Piasecki, S., Nielsen, L. H., & Mathiesen, A. (2017). Palynology and terrestrial ecosystem change of the Middle Triassic to lowermost Jurassic succession of the eastern Danish Basin. *Review of Palaeobotany and Palynology*, 244, 65-95.

For the model, water diffusion in wood has been simplified to only take into account the diameter of vessels (or tracheids) as a parameter of porosity of the material.

Concerning *Pseudomytiloides dubius*, Caswell and Coe (2013) suggest a similar life cycle as *Mulinia lateralis* (opportunistic species, 2 years max, sexual maturity at 2 months) rather than *Mya arenaria* (equilibrium species, 20 years max, sexual maturity at 2 years). For the model, the growth pattern of *P. dubius* was extrapolated from large scale regression analyses conducted on bivalve morphologies (Ridgway Richardson and Austad 2011).

Line 116. The use of oysters to estimate the biomass of *Pseudomytiloides dubius* is flawed in my view, as oysters produce substantially thicker shells than did the inoceramids. Therefore, the weight of an equivalent number of *Pseudomytiloides dubius* specimens to oysters will be less (perhaps considerably so), and so the wood log to which they were attached would have remained floating for longer. A number of publications give the size ranges of *Pseudomytiloides dubius* (e.g. Caswell and Coe, 2013) and other publications give the thickness of inoceramid shells, so I think an estimation of *Pseudomytiloides dubius* biomass could be made.

Thank you for this point we have modified the model to reflect more accurately the growth of *P. dubius*.

Lines 133, 139, 140. See point above about the use of oysters as proxies for inceramids.

We have modified the model to reflect the growth of *P. dubius* more accurately.

Line 147. 'Left hand side'. This is confusing. Compared to what? Wood has no directionality. Presumably, the authors are referring to the relative position on the figure.

Altered to “*Of the figure*”

Line 155. A question here. How is it known that the small logs are complete specimens?

We are convinced they are complete logs because there are crinoid growth around the end of the logs whereas in contrast the small fragments have very small individuals.

Line 168-169. It is very dubious to say that modern oysters are the 'nearest modern equivalent' to *Pseudomytiloides dubius* (see points above). The authors should use literature on the growth rates estimated for that species from the literature.

We have altered the model to reflect the growth of *P. dubius* better (see points above).

Lines 178-182. I fail to see how the decrease in density along the log of *Seirocrinus* specimens proves the colony was attached to the log while it was floating rather than on the seafloor (benthic). Surely differential density of specimens could have occurred on a sunken log too. This is not to say I question a pseudoplanktonic habit for *Seirocrinus*, it is just I don't think the spatial analysis proves the point.

Crinoids have an even distribution in benthic communities, without this significant linear decrease

in density.. Modern submersible data (e.g. <https://oceanexplorer.noaa.gov/oceanos/explorations/ex1811/dailyupdates/nov18/media/nov18-1-800.jpg>) shows an even space distribution for the crinoids even with high energy conditions.

Lines 183-186. I really don't understand the logic in these sentences. Wood logs float passively in the sea. They are not 'driven' like boats, so the analogy of a 'front' and 'back' to a log does not make sense. The 'drivers' for floating logs will be wind and waves, which act across the whole of log, depending for wind on how much of the log projects out of the water. Thus, a log is more likely to be pushed horizontally in the water column, so will not have a 'font' or 'back'.

Logs would travel in a streamlined position within the current, so facing into the current. Therefore, the log structure would have front and back. This log would still being pushed along by currents and wind. The structure will still have an orientation in the path of least resistance. So there is a front and a back but its movement is passive: [Gradient of energy] preferential orientation or direction. We have added the word "*passive*" We have also added that the structure "*would have a preferential orientation*" Lines 215.

Line 187. Could the authors expand on the evidence that the *Seirocrinus* specimens were attached onto the underside of the log.

We agree that this needs further explanation. There are (undeveloped) crinoid attachment discs on the top surface. We have added "*the top side is exposed with either smaller individuals or undeveloped attachment disks*" Lines 218-219.

Line 218. What the process by which *Seirocrinus* specimens seal wood, preventing water ingress? They did not have substantial holdfasts, being instead attached using cirri, so I don't see how they would have provided much in the way of sealing.

Thank you for pointing this out, we have removed this part of the discussion.

Lines 236-238. This goes back to the question from line 155. Another explanation for large specimens on small pieces of wood is that only a small piece of the available wood was collected, i.e. only the bit with the nice specimen. Wood from the Posidonienscheifer is difficult to collect because the surrounding shale tend to flake into tiny pieces.

The evidence does not support this idea: The smaller specimens have individuals with smaller body sizes, while the large specimens fully grown and attached to fragments. So must have broken off the larger logs.

Lines 249-250. These statements have been made before – repetition.

We agree and so have removed these statements.

Line 262. The conclusion section should just be a precis of information and discussion already given in the rest of the paper. This conclusion contains additional thoughts that should be in the discussion, and is therefore not a conclusion.

Thank you for this suggestion. We have integrated these points into the end of the discussion.

Line 265. Given that the studied Toarcian logs had a maximum of two pseudoplanktonic species (Seirocrinus subangularis and Pseudomytiloides dubius) how can these ‘rafts’ be ‘larger’ and ‘more complex’ than modern floating wood communities, which can have many, many more taxa on them, on very large logs.

We agree “*more complex*” has been removed.

Line 266. How was the existence of the Seirocrinus subangularis floating colonies a ‘necessity as a result of a high number of anoxic shallow water basins’? Is the point being made here that during the Toarcian

OAE the benthic mode of life was not available to crinoids in these basins?

This is a good comment we have added “*that meant the benthic mode of life was not available to crinoids in these basins*”

Lines 271-272. What has angiosperm wood got to do with this story, given angiosperms did not evolve until the early Cretaceous? Also, what is the unique ecological phenomenon being referred to? This can't be pseudoplankton, which are just as prevalent today as in the Toarcian.

We have added: “*of pseudoplanktonic crinoids and bivalves rafts colonies had vanished from the fossil record forever*”.

Conclusion: we have demonstrated the mechanisms for how Jurassic Pseudoplanktonic communities existed, showing that this is indeed possible.

Appendix C

11th June 2020

Editorial staff, *Royal Society Open Science*

Dear Editor

On behalf of myself and the authors, please find attached our further revised manuscript entitled *Reconstructing the ecology of a Jurassic pseudoplanktonic megaraft colony*, which we hope you may consider for publication as a research article in *Royal Society Open Science*. We hope we have addressed all the further comments and concerns from the Associate editor and the two reviewers and we have attached a reply to reviewers (also below) as well as a track changes version of the MS.

Thank you in advance for your consideration of our manuscript and best wishes.

Best wishes,

Aaron W. Hunter

On Behalf of David Casenove, Emily G. Mitchell and Celia Mayers

Comments to the Author:

The manuscript has been significantly revised by the authors in line with reviewers' comments. There are now a couple of minor points still to address, which should be fairly easily accomplished.

We have addressed all the comments below. In some cases we believe that our MS has already resolved these issues or these issues are beyond the scope the current MS.

Reviewer comments to Author:

Reviewer: 1

Comments to the Author(s)

I found that this ms has been improved, but still there are some places where you may explain in more details, such as the followings.

Line

62: these rafts they represent one of the....  these rafts represent one of the...

63: [6], They...  [6]. They... (change comma to period)

67: Pseudoplanktonic Hypothesis  Pseudoplanktonic hypothesis

We have made these corrections.

102-:

randomly distributed points can be modelled using homogeneous Poisson process and are found, for example, when larvae attach to....

 Initial settlement of larvae might be explained by Poisson process, but later there should be "struggle for survival" among individuals during growth because some are very closely positioned and could not survive successfully. Therefore, I recommend the authors to comment on this later process, not only talking about the initial settlement. Also the authors are advised to comment if all the crinoids on the log attached to the log at the same time, or gradually the log had gathered individuals at different times. This may change the scenario of the authors' spatial analysis.

This is a very good point from reviewer 1. When organisms are too close together to survive they produce spatial segregation due to the thinning out or dying of ones that are too close. This process is not a given (not all organisms will have this effect in all cases) and whether this struggle happens depends on the space and food requirements of the organisms and the available resources. It also can occur at different life-stages. It is present here, and can be seen in the PCF plot (Fig. 1d). We have added this point into the Figure caption (Lines 467-468)

So we have added a couple sentences (lines 222-2225) to take into account.

> regarding attachment - do we have the sizes? are they approximately the same (one event) or a range of sizes (many).

This is an interesting point. Qualitatively there is very little size variation in the attachment structures, A detailed quantitative analysis would require more data collection, and while interesting would not significantly inform on these results, so we consider it beyond the scope of the current MS.

220-238:

In the "Discussion", authors are advised to state clearly why benthic lifestyle cannot be explained by the spatial analysis. As I comment on the first version, half-buried log could host preferentially attached individuals on the log. Your bare surface on the upper side of the log might be the underside of the log where crinoids could not settle on.

[END]

The upper side of the log is not bare: We state (on line 234) that “the top side is exposed with either smaller individuals or undeveloped attachment disks.”

Reviewer: 2

Comments to the Author(s)

The authors have done a reasonable job amending the ms in relation to my review comments, and, in particular, I am pleased they used *Pseudomytiloides* data in their modelling. They have toned down the hyperbole although I still think the phrase ‘largest in-situ monospecific/deuterostome invertebrate accumulations ever to exist in the Phanerozoic fossil record’ is overwrought and unnecessary (as well as now being even more of a mouthful), as I can think of other similar accumulations, such as oyster reefs and serpulid reefs, both of which can be huge (100 m long and metres thick).

We agree with Reviewer 2, this section has now been removed

One issue that I still initially disagreed with the authors on was this comment on their first ms draft:

Lines 183-186. I really don’t understand the logic in these sentences. Wood logs float passively in the sea. They are not ‘driven’ like boats, so the analogy of a ‘front’ and ‘back’ to a log does not make sense. The ‘drivers’ for floating logs will be wind and waves, which act across the whole of log, depending for wind on how much of the log projects out of the water. Thus, a log is more likely to be pushed horizontally in the water column, so will not have a ‘font’ or ‘back’.

To test this I did a small-scale experiment in my bathroom sink with a matchstick and a small piece of Blu-tack. First, I used just the matchstick, blowing it from one side of the sink to the other to see what orientation it moved in, alternating the starting orientation of the matchstick. The resulting movements were quite random, but, as I suspected, quite often the matchstick moved at right angles to my blowing (simulating wind over water). I then tried the same thing with my cupped hand simulating waves, with similar results. Then I added a small amount of Blu-tack to one end of the matchstick, enough to cause that end of the matchstick to sink slightly underwater and the unweighted end to raise slightly out of the water. This was to simulate the case in the paper of crinoids preferentially attached to one end of the log. When I then simulated wind and waves I found a very interesting thing, and that was regardless of the starting orientation, the matchstick always rotated and then moved with the weighted end trailing (the ‘back’) the unweighted end leading (the ‘front’). I presume this is because the weighted end acts with greater friction in the water. This is exactly what is described in the MS, so I take back my initial

criticism!

We thank reviewer 2 for conducting this small experiment that confirms our findings.